



**Cloud phase estimation and macrophysical properties of low-level clouds using in-situ and radar**
**measurements over the Southern Ocean during the SOCRATES campaign**
Anik Das[1], Baike Xi[1], Xiaojian Zheng[2], and Xiquan Dong[1]*
[1]Department of Hydrology and Atmospheric Sciences, University of Arizona, Tucson, AZ, USA
[2]Environmental Science Division, Argonne National Laboratory, Lemont, IL, USA
**Correspondence:** Xiquan Dong (xdong@arizona.edu)
**Abstract.** The Southern Ocean (SO) provides a unique natural laboratory for studying cloud formation
and cloud-aerosol interactions with minimal anthropogenic influence. The Southern Ocean Clouds,
Radiation, Aerosol Transport Experimental Study (SOCRATES), was an aircraft-based campaign
conducted from January 15 to February 28, 2018, off the coast of Hobart, Tasmania. During
SOCRATES, the NSF/NCAR GV research aircraft, equipped with in-situ probes and remote sensors,
observed aerosol, cloud and precipitation properties, and provided detailed vertical structure of clouds
over the SO, particularly for the low-level clouds (below 3 km). The HIAPER Cloud Radar (HCR) and
in-situ cloud and drizzle probes (CDP and 2DS) measurements were used to provide comprehensive
statistical and phase-relevant macrophysical properties for the low-level clouds sampled by the 15
research flights during SOCRATES. A new method based on HCR reflectivity and spectrum width
gradient was developed to estimate cloud boundaries (cloud-base and -top heights) and classify cloud
types based on their top and base heights. Low-level clouds were found to be the most prevalent, with
an almost 90% occurrence frequency. A new phase determination method was also developed to
identify the single-layered low-level clouds as liquid, ice, and mixed-phases, with occurrence
frequencies of 45.4%, 32.5%, and 22.2%, respectively. Low-level clouds over the SO have significantly
higher SLW concentrations, with liquid being most prevalent at higher temperatures, ice phase
dominating at lower temperatures, and mixed phase being least common due to its thermodynamic
instability. Regarding their vertical distributions, the liquid phase occurs most frequently in the lower
mid-cloud range (from 500 m to 1 km), the mixed phase dominates at cloud bases lower than 1 km but
is well distributed along the vertical cloud layer, while the ice phase is prevalent from the middle to
upper cloud levels (1-3 km). The higher occurrence of the mixed phase at the cloud base could be
attributed to large drizzle-sized drops and/or ice particles.
**1 Introduction**
Southern Ocean (SO) clouds impact the radiation budget over the region in a significant manner (Kay
et al., 2012; McCoy et al., 2014) which the global climate models cannot simulate accurately (Bodas-
Salcedo et al., 2016; Cesana & Chepfer, 2013; Kay et al., 2016; Trenberth & Fasullo, 2010; Wang et
al., 2018), which tends to underestimate the shortwave fluxes, also producing lower cloud fraction and
less supercooled liquid water than observed (D'Alessandro et al. 2021). The SO represents a remote,
pristine, and pre-industrial environment (Hamilton et al., 2014; Uetake et al., 2020; Humphries et al.,
2021) and provides a unique natural laboratory to understand cloud formation and microphysical
properties, cloud-aerosol interactions with minimal anthropogenic influences (McCoy et al., 2015; Xi
et al., 2022).
The low-level SO clouds feature a predominantly high concentration of supercooled liquid water
(SLW, almost 80% of low-level clouds contain SLW over a temperature range of -40 to 0℃, Hu et al.,
2010). Their cloud macrophysical and microphysical properties are different from subtropical marine
boundary layer (MBL) clouds which contain almost all liquid clouds (Dong et al., 2014; Wu et al.,
2020; Zhao et al., 2020) and from the Arctic mixed-phase clouds with a top layer of liquid and bottom
layer of ice clouds (Qiu et al., 2015). Understanding the dominant cloud phase and phase-related spatial
homogeneity of the low-level SO clouds is crucial to expanding our current understanding of the region
along with developing better parametrization for the increased accuracy of the global climate model


predictions (Zhao et al, 2023; Liu et al 2023; etc.). Identifying the cloud phase is crucial to accurately
retrieving cloud macrophysical and microphysical properties because most algorithms are tuned for
specific cloud phases over different climatic regions (Shupe, 2007). Incorrect parametrization of low-
level clouds is a key climate uncertainty and bias; and causes wide intermodel variation (~50%) (Klein
et al., 2017) as liquid-to-ice conversion of cloud particles reduces albedo at the top of the atmosphere
(TOA) (Xi et al., 2022).
Several studies exist on classifying cloud-type, cloud phase and hydrometeor-type detection over
the SO region (e.g., Xi et al., 2022; Desai et al., 2023; D'Alessandro et al., 2021, 2019; Romatschke &
Vivekanandan., 2023; Atlas et al., 2021; Schima et al., 2022; Zaremba et al., 2020) and Arctic clouds
(e.g. Shupe 2007; Korelov & Milbrandt, 2022). They utilized a suite of in-situ, radar-lidar and machine-
learning approaches to predict cloud phase or cloud-hydrometeor types along with their relevant macro-
and micro-physical properties but reported a significant difference in phase retrieval results and phase
transition processes based on the nature of the campaign and instrumentation. These studies have
various performances depending on their retrieval methods and assumptions during retrievals. Xi et al.
(2022) used the W-Band radar measurements and microwave radiometer retrieved cloud liquid water
path (LWP) to estimate cloud phase and macrophysical properties over the SO (North of 60° and South
of 60° latitude) for clouds sampled during the ship-based the Measurements of Aerosols, Radiation, and
Clouds over the Southern Ocean (MARCUS, Xi et al., 2022; Marcovecchio et al., 2023; McFarquhar
et al., 2016, 2021) campaign and estimated a greater frequency of mixed-phase clouds followed by ice
and liquid clouds. Ship-based measurements during the MARCUS can provide accurate cloud
boundaries and their vertical distributions. Wang & Sassen (2001) presented algorithms for retrieving
cloud macrophysical properties, such as boundary, thickness, phase, type, and precipitation, using a
combination of ground-based lidar, millimeter-wave radar, IR radiometer, and MWR measurements at
the ARM SGP CART site in Northern Oklahoma. Further, Shupe (2007) provided an array of ground-
based Lidar-Radar threshold values to estimate cloud hydrometeor phase including aerosols, liquid,
mix, ice, drizzle and rain designed for the study of Arctic clouds. Compared to the ground-based
measurements, the aircraft in situ measurements, however, can provide more reliable datasets without
the issues of retrieval methods and assumptions because aircraft can fly in greater proximity to the cloud
boundaries and even inside the cloud layers. Also, the onboard radar and lidar suffer less attenuation
than the ground-based remote sensors (Ewald et al., 2021).
The Southern Ocean Clouds, Radiation, Aerosol Transport Experimental Study (SOCRATES)
aircraft field campaign provided a valuable dataset to investigate the MBL clouds over the SO. The
SOCRATES was an aircraft-based campaign that used the National Science Foundation (NSF)/National
Center for Atmospheric Research (NCAR) Gulfstream-V (GV) research aircrafts based out of the coast
of Hobart, Tasmania (42-62°S and from 133°-163°W) from 15 January to 28 February 2018, targeting
cold sector boundary layer clouds and airborne sampling of in-, below- and above-cloud transects
obtaining both time series and vertical cloud information using an array of in-situ cloud and drizzle
sampling probes and radar-lidar instruments, mostly spanning a period of midnight to early morning
for each flight track on subsequent days. The in-situ probes and remote sensors (cloud lidar and radar)
onboard the aircraft flown during the SOCRATES campaign provide a direct observation of
precipitation, cloud particles, and aerosols below, inside and above the cloud layers sampled, along
with vertical profiles, for a better characterization of the MBL structure and free troposphere.
D'Alessandro (2021) used the suite of in-situ cloud and drizzle sampling probes (CDP & 2DS) onboard
the NCAR-GV aircraft during SOCRATES to estimate cloud phase heterogeneity and frequency
distributions predicting significant SLW and ice phase concentrations using a multinomial logistical
regression model (MLR). Romatschke and Vivekanandan. (2023) used a fuzzy logic scheme to classify
cloud hydrometeor type as a time-height profile using an array of cloud radar-lidar derived values.
According to Wang et al. (2012), integrating in-situ sampling capabilities with remote sensing
measurements offers significant advantages for studying atmospheric processes. In this context, the
integrated 2-dimensional cloud profiles obtained through remote sensing of microphysical processes





complement the detailed size-resolved distributions captured by in-situ cloud measurements. Therefore,
solely relying on either in-situ or remote sensing measurements can lead to certain disagreements in
cloud profile as the sampling probes can only detect cloud particles at the flying altitude while the
remote sensing profiles can provide vertically resolved cloud profile but with an offset of around 100-
200 meters. The lidars have a smaller operating wavelength compared to radar and provides well-
resolved vertical profiles for detecting aerosols, optically thin clouds and cloud boundaries, (Wang et
al., 2012, 2009; McGill et al., 2002) but its signals are easily attenuated by optically thick clouds, such
as liquid clouds (Sassen., 1991) as observed over the SO. Therefore, we exclusively used radar
measurements to estimate cloud boundaries and cloud phase for optically thick clouds in this study.
Furthermore, we also tune the High Spectral Resolution Lidar (HSRL) measured Particle
Depolarization Ratio (PLDR) thresholds based on the phase estimation method presented in this study.
This adjustment was seen necessary because the existing PLDR thresholds presented in Sassen (1991),
Intrieri (2002), and Shupe (2007) were developed for Arctic clouds, which differ significantly from the
low-level clouds over the Southern Ocean (SO).
In this study, we aim to use a combination of both in-situ and radar-based measurements during
SOCRATES to develop a new method to classify the MBL cloud phase and determine cloud boundaries
over the SO for low-level clouds. The newly developed classification method can be used to help answer
the following scientific questions:
1.  What are the dominant cloud types, their associated cloud phase, base and top heights, and their
vertical distribution?

2.  What are the phase-specific macrophysical properties for SO low-level clouds sampled during
the SOCRATES campaign?

The paper is organized in the following manner: data and methods are introduced in Section 2. The
statistical results for all cloud properties during the SOCRATES campaign are presented in Section 3.
Cloud phase-specific results and comparisons with other algorithms for the low-level clouds are
discussed in Section 4, and finally, Conclusions and Summary are given in Section 5.
**2 Data and methods**
**2.1 SOCRATES aircraft campaign**
The Southern Ocean Clouds, Radiation, Aerosol Transport Experimental Study (SOCRATES) aircraft
field campaign was conducted over the SO with a total of 15 research flights from 15 January to 28
February 2018 (McFarquhar et al., 2021). During SOCRATES field campaign, the Cloud Droplet Probe
(CDP) and 2-dimensional stereo- particle imaging probe (2DS) were utilized to measure cloud and
drizzle microphysical properties, respectively. Additionally, HCR and HSRL were installed on the RV
aircraft to detect cloud structure, phase and boundaries (McFarquhar et al., 2021). HSRL particle linear
depolarization ratios (PLDR) were widely used as a screening tool for cloud phase determination with
liquid clouds having PLDR less than 0.11, mixed-phase clouds falling between 0.11 and 0.15, and ice
clouds having PLDRs greater than 0.15 (Shupe et al., 2005; Xi et al., 2022). The 2DS in-situ
measurements serve as an additional screening to eliminate the ice particles (D>200 µm). More
instrumental details about the SOCRATES campaign can be found in McFarquhar et al. (2021).
The suite of in-situ probes and radar-lidar instruments onboard the SOCRATES aircraft is listed in
Table 1, along with their detection limits and uncertainties. The particle size distribution and number
concentration were retrieved from the CDP and 2DS microphysical probe measurements and merged



to create one continuous dataset with size bins from 2 to 40 µm corresponding to cloud droplets and 40
µm above for drizzle particles (to 1280 µm), at a 1 Hz temporal resolution for each research-flight.
Reflectivity (dBZ), Doppler Velocity ($V_d$) (m/s), and Spectrum Width (WID) (m/s) were retrieved from
the original 2 Hz temporal resolution of the HCR radar measurements and were averaged to match the
1 Hz frequency resolution of the in-situ probes and further interpolated to fixed radar-heights at a range
gate of 19.2 meters. The same treatment was done for the HSRL-retrieved Backscatter Coefficient β
($m^{-1}sr^{-1}$) and Particle Linear Depolarization Ratio (PLDR) (unitless). The HSRL (lidar) signal is highly
sensitive to greater cloud droplet concentrations and can be attenuated within a few hundred meters in
liquid cloud layers (Ewald et al., 2021; Sassen, 1991). Thus, it is not used for phase or boundary
estimation in optically thicker MBL clouds discussed in this paper. The atmospheric temperature (°C)
for the cloud samples was retrieved from the 2-dimensional ERA5 reanalysis product, which is available
in the HCR-HSRL merged dataset at the EOL data archive. This dataset matches the vertical and
temporal resolution of the HCR-HSRL data (NCAR/EOL HCR Team., 2023). Temperatures below -40
oC are not considered during further analysis as they represent homogenous freezing temperatures
(majorly all ice) and most mixed phases exist only over the range of -40 to 0 °C (e.g., Shupe et al.,
2007). The dataset is further filtered to keep only the nadir or zenith pointing direction of the HCR-
HSRL merged dataset, all the in-transition or rotational pointing directions (which were not equal to
±90 degrees) were removed. The 2DS particle morphology or habit imagery data (Wu and McFarquhar.,
2019) was also retrieved and visualized using the Illinois/ Oklahoma Optical Probe Processing Software
(XPMS2D, UIOOPS, McFarquhar et al., 2018).
**Table 1. Measurements from specific instruments used in this study and their relevant properties.**

| | INSTRUMENT | MEASUREMENT | Size Range/Resolution | Uncertainties | REFERENCES |
|---|---|---|---|---|---|
| **In-Situ Probes for Bulk Cloud Sampling** | Cloud Droplet Probe **(CDP)** | Size distribution and concentration of hydrometeors with a diameter between 2-50 µm | 2-50 µm | Cannot resolve non-spherical particles accurately | Lance et al., 2010 |
| | Two-Dimensional, Stereo, Particle Imaging Probe **(2D-S)** | Size distribution and concentration of hydrometeors with a diameter between 10 to 1280 µm range | 10-µm<br><br>D>40 µm for all particles<br><br>D> 200 µm for ice | Cannot resolve for particle sizes D <50 µm, also ice particle detection is certain only for D>200 µm. | Lawson et al., 2006; 2008<br><br>Baker et al., 2009<br><br>Wu and McFarquhar., 2019 |
| **Radar-Lidar remote sensors** | HIAPER Cloud Radar **(HCR)** | Reflectivity, Doppler Velocity, Spectral Width, Linear Depolarization Ratio (LDR), etc. | ~19 m in vertical resolution<br><br>Frequency: 94.40 GHz | Attenuates for larger particle sizes | NCAR/EOL HCR Team., 2014<br><br>Romatschke et al., 2021<br><br>Vivekanandan et al., 2015 |
| | High Spectral Resolution Lidar **(HSRL)** | Backscatter Coefficient, Particle Linear Depolarization Ratio (PLDR), Extinction Coefficient, etc. | Wavelength: 532 nm | Sensitive to optically thin cloud layers | NCAR/EOL HSRL Team., 2012<br><br>Eloranta., 2005 |





## 2.2 Estimation of cloud boundaries

There are multiple existing methods of estimating cloud base and top heights, for example, using thresholds for lidar returned power, depolarization, or backscatter (e.g., Intrieri et al., 2002; Kang et al., 2021, 2024) or thresholding in-situ measured vertically resolved liquid water path (LWP) or liquid water content (LWC) and cloud droplet number concentration. For this study, the cloud base was estimated as the lowest height (from the sea surface) where the HCR Spectrum Width (WID) Gradient is the lowest in value (or highest negative gradient). The lowest WID gradient indicates the change from a precipitation layer to the cloud layer where the gradient of spectrum width decreases sharply. The cloud-base height ($H_{base}$) was also estimated using the HSRL backscatter coefficient threshold, but this cloud-base height was found to be around 400 m higher than that derived from HCR spectrum width gradient. Higher spectrum width around the cloud base indicates a greater turbulence and wider range of particle velocities observed which correlate to potentially stronger turbulence, and likely drizzle or precipitation. This argument is further improved by the aircraft in situ measured microphysical properties.

The cloud-top height ($H_{top}$) is measured as the highest height (from the cloud base) where prominent HCR reflectivity (dBZ >-50) is observed following the method of Kang et al. (2024). Finally, cloud thickness ($\Delta H$) is estimated as the difference between cloud-top and -base heights, $\Delta H = H_{top} - H_{base}$. For double-layered clouds, each single-layer cloud-base and -top heights were identified separately but are not reported in this study as only single-layered cloud types were used for further analysis. The accuracy of the estimated cloud-top and -base heights for the low clouds have been verified by mapping them on the radar detected cloud profile (Fig. 1.) and will be discussed in following sections. This method of estimating $H_{top}$ and $H_{base}$ can be used to find cloud boundaries in the absence of a readily available radiosonde or dropsonde measured cloud heights dataset.

A case study for the retrieved cloud boundaries using the HCR reflectivity and spectrum width gradient is illustrated in Fig. 1(a-d). Using the spectrum width gradient for estimating cloud-base heights allows finding cloud and drizzle base above the precipitating layer with minimal errors compared to lidar HSRL-Backscatter values which attenuate faster for thicker cloud layers. Boundary estimation was carried out for each sublayer separately for double-layered clouds (not shown). Isolated cloud transects less than 50 meters in vertical height and 10 seconds in temporal width (which appear as small, isolated dots or patches of reflectivity or spectrum width profiles in the following plot) are ignored as noise.

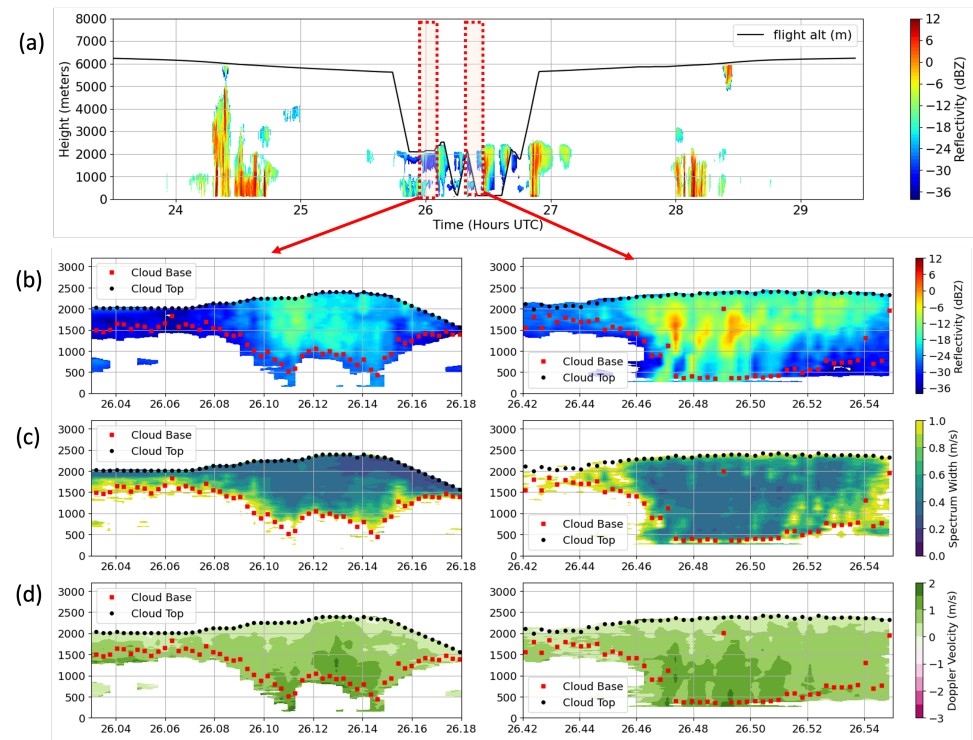

**Figure 1. (a) HCR Reflectivity (dBZ) profile for the entire flight track for RF01 from 23:30 UTC (15 Jan 2018) to 05:30 UTC (16 Jan 2018). The solid black line in (a) indicates the flight altitude in meters. The two red boxes in (a) are the subsection for which the cloud-top and -base heights are displayed (b), (c), and (d). The left panels represent the cloud profiles when the flight was flying above the cloud top and radar pointed nadir, while the right panel shows a zenith-pointing radar cross-section with the flight flying below the cloud base. (b), (c), and (d) are the profiles of HCR reflectivity, spectrum width (WID) and Doppler Velocity ($V_d$) with the cloud tops (black squares) and cloud bases (red squares).**

**Table 2. Cloud classification using Base and Top heights.**

| CLOUD TYPE | CLASSIFICATION METHOD |
|---|---|
| LOW (low-level clouds) | $H_{top} \leq 3km$, in a single layer |
| MID (middle level) | $H_{base} > 3km$ and $H_{top} \leq 6km$, in a single layer |
| HGH (high clouds) | $H_{base} > 6km$, in a single layer |
| MOL (mid-over-low) | $H_{base} < 3km$ and $H_{top} \leq 6km$, may not be single layer |
| HOM (high-over-middle) | $3km < H_{base} < 6km$, and $H_{top} > 6km$, may not be single layer |
| HML (high-over mid and low) | $H_{base} < 3km$, and $H_{top} \geq 6km$, extend over the whole tropospheric layer |



## 2.3 Classification of Cloud type

After estimating the cloud-base, -top heights, and cloud thickness ($H_{base}$, $H_{top}$, $\Delta H$), the cloud types are categorized following a classification method described by Xi et al., 2010, and summarized in Table 2. A single layer means that there is no other cloud layer above or below the classified cloud layer for the time series.

Based on this classification method, a significant number of LOW ($H_{top} \leq 3km$) clouds were identified followed by MOL ($H_{base} < 3km$, $H_{top} \leq 6km$) and MID ($H_{base} > 3km$, $H_{top} \leq 6km$). Some HGH ($H_{base} > 6km$), HOM ($3km < H_{base} < 6km$, and $H_{top} > 6km$), and HML ($H_{base} < 3km$, and $H_{top} \geq 6km$) cloud types were also identified but almost insignificant or negligible in number relative to LOW, MID, and MOL types. Only a couple of flights had single or double-layered clouds with $H_{base}$ and/or $H_{top} > 6km$, which were ignored to optimize for statistical deviation and minimize errors. Notice that these results are due to the selected cloud cases during the flight, which may not represent all the true cloud types.

Cloud phase estimation was only carried out for the single-layered LOW cloud type, which was the predominant cloud type during the SOCRATES field campaign. The statistical results for the predominant cloud types of LOW, MID and MOL are further discussed in Section 3.

## 2.4 In-Cloud Conditions

The cloud-droplet number concentration and particle size from the merged CDP+2DS dataset is used to calculate a continuous liquid water content (LWC) in $g/m^3$ for cloud and drizzle particles, using the equation (Kang et al., 2021; Zheng et al., 2024) as follows:

$$\text{LWC} = \frac{4}{3} \pi \rho_w \sum_{\{i=1\}}^{\{n\}} r_i^3 . N_i, \qquad (1)$$

where $\rho_w$ is the density of liquid water, $r_i$ is the particle radius measured as droplet size distribution from the CDP+2DS particle size bins, and $N_i$ is the number concentration ($\#/cm^3$) per bin. The LWC values are further used to compute the liquid water path (LWP) in $g/m^2$ as a function of cloud thickness ($\Delta H$) (Oh et al., 2018), as follows:

$$LWP = \sum_{\{j=1\}}^{\{n\}} LWC_j . \Delta H_j. \qquad (2)$$

The in-cloud conditions were constrained to keep cloud samples only containing LWC greater than 0.001 $g/m^3$, to remove noise or uncertainty in measurements. CDP number concentration less than 1 $cm^{-3}$ corresponds to ice phase observations in particles with size less than 50 μm, and greater than or equal to 1 $cm^{-3}$ corresponds to the liquid phase (D'Alessandro et al., 2021). Hence, choosing LWC threshold is based on the decision to ensure a significant cloud density and cloud number concentration, and remove clear-sky conditions or noise from aerosols. The number concentration of ice particles with diameters greater than 200 μm is very low (Zheng et al., 2024), as shown in Fig. 2, indicating that most ice phase particles occur below the 2DS-defined threshold of 200 μm for ice particle size distribution (Wu and McFarquhar, 2019).

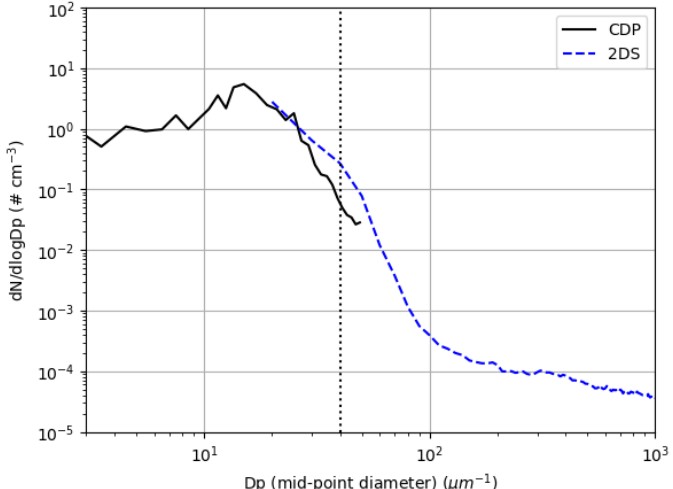

**Figure 2. Cloud and Drizzle (CDP+2DS) size distributions aggregated for the 15 research flights during SOCRATES. The dashed line at $D_p = 40$ μm denotes the separation from cloud to drizzle droplets.**

There is a small offset in the calculated LWP because LWC is derived from in-situ cloud (CDP) microphysical properties while cloud thickness is derived from the HCR and HSRL measurements. However, the difference was insignificant when these results were compared to the average LWP values observed over the SO by previous studies like Xi et al. (2022) and Mace et al. (2021). Future work could be done on finding the LWC and LWP as a function of the cloud vertical height profiles matched to the HCR-HSRL profiles, following the methods mentioned in Vivekanandan et al. (2020). Ice water content (IWC) in g/m$^3$ was also retrieved from the 2DS dataset for particle sizes $\geq$ 200 μm. The 2-dimensional HCR-HSRL parameters were further constrained for the cloud base and top heights for the classified cloud types in the vertical dimension and for the LWC threshold in the time dimension. The 2-dimensional air temperature ($^{o}$C) from ERA5 was filtered to extract the cloud-base and -top temperatures. The aircraft-measured air temperature is not used in this study as it only measures temperature at the fixed flying altitude of the aircraft.

### 3 Statistical results for prominent cloud types

The most prominent cloud types identified using the cloud boundary estimation discussed in the Section 2.3, such as LOW, MID, and MOL, which are consistent of ~ 85-90% occurring frequencies in total from the 15 research flights during SOCRATES. Figure 3 summarizes the occurrence frequencies of the classified single-layered LOW, MID, and MOL clouds along with their vertical structures or the thickness (in km). Multilayered clouds were not considered for this classification due to their negligible occurrence frequencies compared to the single-layered clouds. LOW clouds are the most observed cloud type (~90%) compared to the other two cloud types (less than 10%), due to the nature of sampling and targeted cloud sector of the SOCRATES campaign. Most of the research flights flew below 6-7 km over the SO studying MBL clouds with greater amount of SLW (Schima et al., 2022).

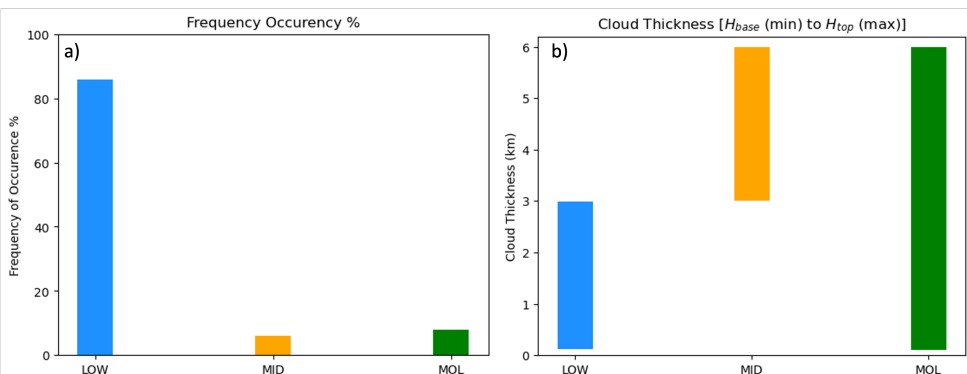

**Figure 3. (a) The occurrence frequencies of single-layered LOW, MID, and MOL clouds, b) the**
**average thickness for each cloud type.**
Figure 4 visualizes the LWP frequencies for each cloud type averaged from the 15 research flights
along with the constrained LWP frequency which shows the percentage occurrence of LWP above the
threshold of 10 g/m$^2$ for each cloud type. Amongst the 15 flights, the maximum average LWP is
observed for MOL clouds at around 370 g/m$^2$ followed by LOW at 208 g/m$^2$ and very low for MID
clouds at around 65.8 g/m$^2$. The overall statistical results for the LOW, MID, and MOL classified
cloud types are summarized in Table 3.

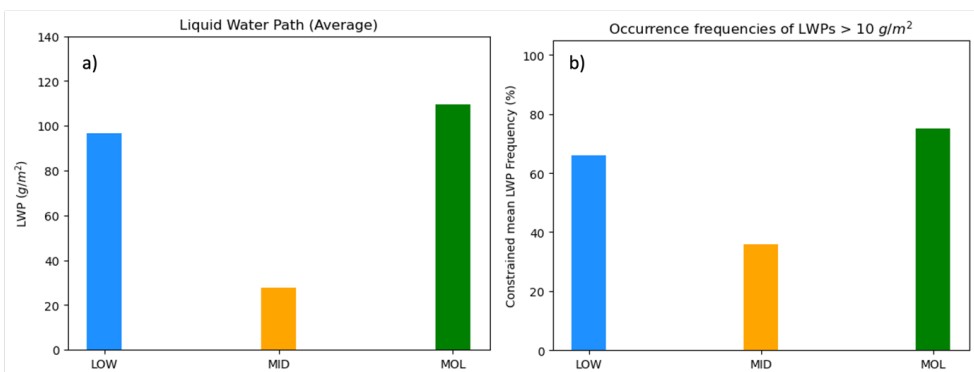

**Figure 4. a) Averaged LWP for each cloud type from the 15 research flights. b) Constrained LWP**
**occurrence frequencies for LWPs greater than 10 g/m$^2$ which is the threshold for classifying liquid**
**and mix phase from ice-phase cloud transects. Percentages for LWP>10 g/m$^2$ are approximately**
**66%, 36% and 75% for LOW, MID and MOL clouds, respectively.**







**Table 3. Mean, standard deviation, minimum and maximum ranges for the estimated cloud base**
**and top heights along with the calculated LWP values for each single-layered cloud type.**

|  | LOW | MID | MOL |
|---|---|---|---|
| $H_{base} \pm$ SD | $1.01 \pm 0.6$ | $4.16 \pm 0.80$ | $1.56 \pm 0.93$ |
| Min, Max (km) | 0.13, 2.97 | 3.01, 5.97 | 0.11, 2.99 |
| $H_{top} \pm$ SD | $1.57 \pm 0.58$ | $4.79 \pm 0.84$ | $4.18 \pm 1.04$ |
| Min, Max (km) | 0.15, 2.99 | 3.03, 5.99 | 3.01, 5.99 |
| LWP (mean) $\pm$ SD | $96.7 \pm 187$ | $27.9 \pm 73$ | $109.48 \pm 208$ |
| Max ($g/m^2$) | 2732 | 963 | 2121 |

**4 Low cloud phase retrieval results and discussions**
As shown in Figure 3, LOW clouds are dominant cloud type (~90%) observed during SOCRATES. In
this section, we discuss how to determine LOW cloud phase. The estimated cloud base and top heights,
along with other radar-lidar and in-situ variables, were aggregated using the median to a temporal
resolution of 10 seconds (0.1 Hz). This aggregation was constrained to the cloud boundaries and single-
layered low-level cloud types to ensure a more continuous data distribution and to minimize outliers,
thereby improving statistical consistency before phase determination.
**4.1 Determination of Cloud Phase**
Figure 5 describes the flow-chart of determining cloud phase for the classified low-level clouds (LOW)
with cloud-top heights below 3 km after constraining for the in-cloud conditions (LWC>0.001 $g/m^3$).
The phase partitioning method described in this section is used simultaneously as combined filters to
classify the cloud phase as a 2-dimensional phase profile of liquid, mix and ice phase. LWP threshold
was estimated after constructing probability density function (PDF) plots for the classified LOW, MID,
MOL, HGH, and HOM clouds which returned a peak of less than 10 $g/m^2$ in LWP values for the HGH
and HOM clouds which are prevalently ice-dominated clouds. There are significant overlaps between
the LWP PDFs for classified cloud types, which results in some inconsistencies and uncertainties. For
example, LOW clouds also display a peak in LWP values less than 10 $g/m^2$ but simultaneously also
show consistently greater frequency for LWP values even greater than 200 $g/m^2$. Whereas the LWP
frequencies for other cloud types diminish to zero after 15 or 20 $g/m^2$. The LWP value of ~10 $g/m^2$ also
lies around the uncertainty value for LWP measurement over SO (Kang et al., 2021). While LWP values
for LOW, MID and MOL cloud types are significantly higher, most of the LWPs for HGH and HOM
clouds are below 10 $g/m^2$. Therefore, we use LWP=10 $g/m^2$ as a threshold to determine cloud phases
where cloud samples with LWP < 10 $g/m^2$ is classified as ice clouds, while LWP >= 10 $g/m^2$ as mixed-
phase or liquid clouds.



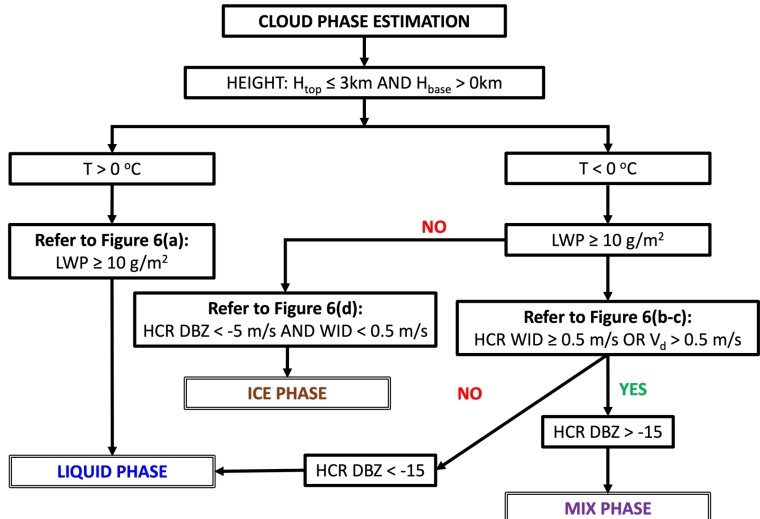

**Figure 5. Flow chart depicting the phase classification of single-layered LOW clouds during**
**SOCRATES. Spectrum Width (WID), Doppler Velocities ($V_d$) and reflectivity (dBZ) are**
**measured by HCR radar, and Liquid Water Path (LWP) is calculated from in-situ measurements.**
**Temperature is provided from ERA5 reanalysis air temperature product matched to the HCR-**
**HSRL merged dataset.**

In addition to the LWP threshold, the profile of atmospheric temperature provides another threshold
for determining LOW cloud phase. where temperature is greater than 0 °C is classified as liquid while
for temperature between -40 to 0 °C is further analyzed for categorizing between liquid, mixed and ice.
As shown in Fig. 5, if LWPs are equal or greater than 10 g/m$^2$ and T is higher than 0 °C, then the cloud
samples are classified as liquid clouds. For ice clouds, it is as simple as classifying liquid clouds where
the cloud samples are defined ice clouds when both LWP < 10 g/m$^2$ and T < 0 °C, whereas it becomes
more complicated for mixed-phase clouds. For the cloud samples with T > 0°C and reflectivity < -15
dBZ are considered as liquid cloud droplets because most of drizzle drops have higher reflectivity (> -
15 dBZ, Wu et al., 2020). Low WIDs correspond to homogenous single-phase of cloud hydrometeors
while a broad range of WIDs suggest multiple phases and/or significant turbulence and wind shear
(Shupe., 2007). The regions with both low WID and $V_d$ (< 0.5 m/s, weak turbulence) are classified as
liquid phase with dominant small liquid cloud droplets and SLW. the cloud samples are classified as
mixed-phase clouds when both Doppler spectrum width (WID) and Doppler velocity ($V_d$) values are
greater than 0.5 m/s (downdraft) which represent a greater variability of velocity, greater turbulence in
cloud droplets and variable size distribution including large ice or drizzle-sized particles. The cloud
samples with either WID > 0.5 m/s and $V_d$ < 0.5 m/s (or WID < 0.5 m/s and $V_d$ > 0.5 m/s) i.e. regions
of high WID but low $V_d$ (updrafts) or low WID and high $V_d$ (downdrafts) are re-classified into mixed
phase if the reflectivity (dBZ) > -15 and liquid phase if the reflectivity (dBZ) < -15. Radar reflectivity
is a function of the sixth moment of the particle size (Wang et al., 2009) hence small and uniform liquid
cloud droplets exhibit significantly lower reflectivity while mixed-phase clouds normally exhibit higher
reflectivity values with a higher in-homogenous particle size distribution and greater density variability.
Furthermore, the cloud samples with reflectivity > 5 dBZ represent precipitation (rain or snow
depending on the temperature) and hence these samples are omitted in this study. Furthermore, the
cloud samples with reflectivity > 5 dBZ represent precipitation (rain or snow depending on the
temperature) and hence these samples are omitted in this study.



The estimation of WID, $V_d$, and dBZ thresholds was determined by the average values observed for
each cloud layer based on the tropospheric height and at the estimated cloud base. These values were
aggregated by prioritizing regions with more measurements over those with less ones, along with
considering the cloud density at each layer, further comparing with existing studies, such as Xi et al.
(2022) and Shupe (2007). Although these constraints were specifically tuned for clouds sampled during
the SOCRATES campaign, we expect them to be broadly applicable to MBL clouds over the Southern
Ocean.
To further demonstrate our phase classification methodology using T, LWP, WID, $V_d$, and the
reflectivity (dBZ) as shown in Fig. 5, we present the bivariate histograms of liquid, mixed-phase and
ice clouds in Fig. 6. Figure 6a illustrates the liquid cloud droplets, drizzle and rain drops based on radar
reflectivity (-15 dBZ and 5 dBZ) and Doppler velocity ($V_d$<0.5 m/s) at higher temperatures (T>0 °C).
As shown in Fig. 6a, drizzle drops are dominant in the liquid clouds at T>0 °C, with higher radar
reflectivity and Doppler velocity. Marcovecchio et al., 2024 found that there is a higher drizzle
frequency rate (71.8%) over SO using the ship-based radar-lidar measurements during the Measurement
of Aerosols, Radiation, and CloUds over the Southern Ocean (MARCUS) field campaign than the
ground-based radar-lidar measurements at the ARM East North Atlantic (ENA) site (45.1%). Figures
6b-6c present the classification of mixed-phase and liquid clouds where WID ≥ 0.5 or < 0.5 m/s, with
T < 0 °C and LWP ≥ 10 g/m2, using $V_d$ and dBZ thresholds, where higher dBZ (>-15) corresponds to
mixed while lower dBZ (<-15) and lower $V_d$ (<0.5 m/s) corresponds to liquid phase. Liquid droplets
due to their smaller size and uniform homogenous distribution exhibit lower dBZ, and lower $V_d$
(updraft). Higher WID in liquid droplets mostly represent regions of turbulence and wind shear.   The
2D pattern in Fig. 6c mimics that of Figure 6a, indicating that liquid clouds with more drizzle and
mixed-phase clouds are dominant for LOW clouds over SO. There is a linear relationship between cloud
reflectivity (dBZ) and Doppler velocity ($V_d$) for ice clouds, similar to liquid and mixed-phase clouds,
except for Fig. 6b. Figure 6(d) represents regions classified as ice based on lower temperatures (T<0
°C) and LWP < 10 g/m$^2$. The results of Figs. 6c and 6d suggest that the ice particle size distributions
are not as broader as expected even though their particle sizes are much larger with higher radar
reflectivity and Doppler velocity. The threshold of very high reflectivity, dBZ > 5 represents regions of
precipitation (rain or snow) and is not considered in the final phase classification.
The phase classification methodology illustrated in Figs. 5 and 6(a-d) are used together
simultaneously for estimating the liquid, mixed and ice phases for the low-level clouds in this study.
The described method using radar-retrieved and in-situ measurements in this study was compared with
the similar thresholding values defined in previous studies (Xi et al., 2022; Romatschke &
Vivekanandan., 2023; Desai et al., 2023; Wu et al., 2020; Shupe., 2007) for coherence and consistency
in the phase retrieval methodology. There could be some cases where the specified $V_d$ and the WID
threshold yield in a mix or liquid-phase conditions even if the true dominant phase is ice at that level
(Shupe, 2007).
A 2-dimensional cloud phase is determined as a time-height dimensional profile for the valid cloud
segments based on the discussed phase estimation method. Furthermore, the 2D phase profile is used
to find a 1-dimensional dominant phase profile along the time dimension, where the dominant phase is
determined by finding the sample that has the highest sample count along any vertical transect. For
example, if at any time interval the sample count of a particular phase (say Liquid) is greater than the
sample count of the other two phases (Ice and Mix) along the vertical transect, then liquid is the
dominant phase at that time interval, i.e. the phase with the majority sample count at a vertical column
is the dominant phase at that instance. Furthermore, if the sample count of ice is equal to liquid along
any vertical transect, then the dominant phase at that time interval is mixed phase. This 1-dimensional
dominant phase with a temporal resolution of 10 seconds phase partitioning will be used to determine
the phase-specific cloud macrophysical results in the next section.



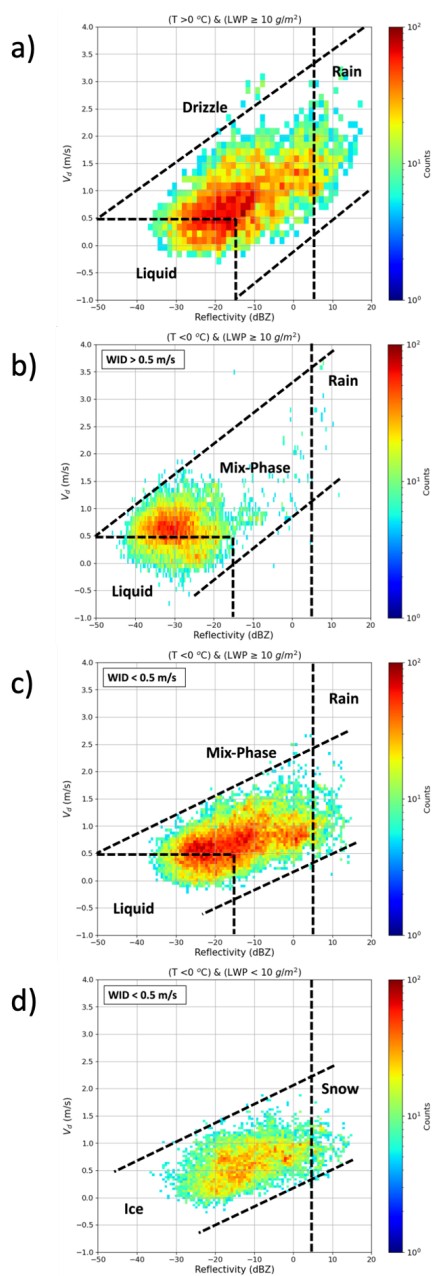


**Figure 6. (a-d) The bivariate histograms of radar reflectivity (dBZ) and Doppler velocity (V$_d$) for**
**different spectrum widths (WID), LWPs and Temperatures (T) to demonstrate the classified**
**liquid, ice and mixed phase cloud samples. The colorbar shows the sample count in each bin in a**
**logarithmic scale and the dashed lines represent the threshold values for the phase classification.**
**It is to be noted that the categories rain, drizzle, and snow are not taken into consideration in this**
**study.**



## 4.2 Cloud Phase Determination Results

The liquid-phase clouds are firstly determined where both temperature (T) is greater than 0 ºC and cloud LWP is greater than 10 g/m$^2$, while when T is lower than 0 ºC and cloud LWP is greater than 10 g/m$^2$, the SLW clouds are determined when both WID and $V_d$ are less than 0.5 m/s. Large ice particles are much heavier than small liquid cloud droplets with a broader spectrum width and greater fall speeds (Xi et al., 2022). The dependence on a linear LWP calculation as per the phase algorithm in this study adds some difficulty in resolving the exact hydrometeor phase in a vertical column. For instance, there could be cloud layers with an ice-phase top and liquid or mix-dominated base, but the LWP constraint considers the whole column to be ice-phase if the LWP < 10 g/m$^2$. A significant number of mixed-phase cloud samples were found at the cloud base due to broader WID and larger $V_d$ values, which could be attributed to the presence of either larger drizzle drops or ice particles.

These estimated phase retrievals are illustrated in Fig. 7(a-h) from one selected case, which was chosen arbitrarily to offer visual clarity in phase profiles. The cloud phase presented in Fig. 7g is the 2-dimensional phase retrieval method but may be not highly depictive of the actual cloud phase, whereas Fig. 7h is the dominant cloud phase for each vertical transect retrieved from the 2D phase data where a phase is considered dominant if its sample count is greater than the other two in the same vertical column. This dominant cloud phase inferred from this 2D data along the vertical axis returns reasonably accurate findings compared to other phase detection studies over the SO. Note that the cloud phase is not available for very low LWC values due to the constraint used for in-cloud conditions.

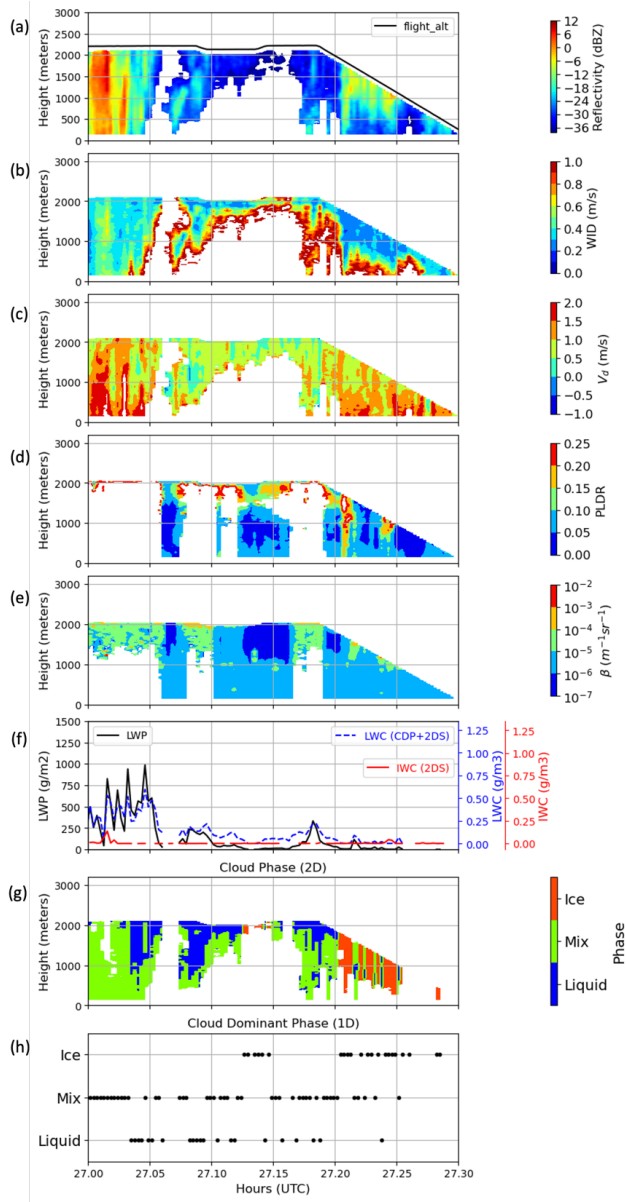

**Figure 7. A case study for flight RF09 during SOCRATES illustrating the phase-detection algorithm in this study. (a) The HCR Reflectivity (dBZ) with the flight altitude in meters (black line), (b) Spectrum Width (WID), and (c) Doppler Velocity ($V_d$) profiles. (d) and (e) represent the HSRL (lidar) Particle Depolarization Ratio (PLDR) and Backscatter Coefficient (β). (f) represents the LWP, LWC, IWC values for each determined phase in (d). (g) and (h) represent the determined 2-D and dominant cloud phase. The time series is in decimal points where 27 hours is 3:00 UTC.**





Higher IWC values correspond to ice phase particles greater than 200 μm in size. It is noticeable
that the ice phase also exists for very low or negligible IWC values which correlates to very small-sized
ice particles. The cloud transects where the dominant phase is liquid but also has a significant amount
of mixed phase around the cloud base is mostly indicative of drizzle or precipitation-size particles.
Inspecting the 2DS particle probe imagery (not shown in this paper) reveals that liquid cloud
droplets are mostly present in the form of spherical shape, and large ice particles have irregular shape,
while small ice particles cannot be resolved very well using the 2DS probes. The 2DS images
demonstrate that liquid cloud droplets are dominant at the upper levels of cloud layer, while a mixture
of liquid cloud droplets and ice particles exists at the lower levels. Large ice particles ($D_p > 50$ μm) are
easily identified by 2DS images, while it is challenging to distinguish small ice particles with cloud
droplets from 2DS imagery. D'Alessandro et al. (2021) developed a phase-determination method by
visually inspecting the 2DS particle imagery for particles of size greater than 50 μm and feeding this
training data to a multinomial logistic regression (MLR) model to classify them as liquid, mix or ice
phase. For the particles of size smaller than 50 μm, they were classified using a simple CDP number
concentration thresholding method: $N_c < 1$ cm$^{-3}$ corresponds to ice phase and $N_c > 1$ cm$^{-3}$ represents
liquid phase.
Table 4a lists the comparison of the phase determination using the MLR method (D'Alessandro et
al., 2022; 2021) and this study. With a total of 2335 overlapping samples, there are 45.7%, 26.2% and
28.0% of classified liquid, mixed-phase and ice clouds from this study, while they are 80.4%, 11.6%
and 7.9% from the MLR method. This comparison indicates that more liquid, but less mixed-phase and
ice cloud are identified by the MLR method than our results for the overlapping samples. Of the three
categories, there are a total of 995 samples of liquid clouds are identified by both the MLR method and
our study at the same timestamps, which accounts for 93.3% of classified liquid clouds from this study
and 53.0% of classified liquid clouds from the MLR method. The overlaps in ice and mixed-phase
clouds from these two methods are much less than their liquid cloud counterparts. The 162 (140)
overlapping samples for ice (mix) correspond to 87% (51%) of classified ice (mix) cloud samples from
MLR method and 24.7% (22.83%) of classified ice (mix) samples detected by this study. Note that
these percentages are just based on the matched dataset samples and do not represent the entire dataset
for both MLR, and the dominant cloud phase determined by this study.
To further evaluate the cloud phase partitioning method, we compare the classified phases from this
study with the MLR method and Shupe et al. (2005) and Intrieri et al. (2002) method for the classified
low-level cloud samples for the 15 research flights during SOCRATES. Shupe et al. (2005) and Intrieri
et al. (2002) used the lidar median PLDR (particle depolarization ratio) values to classify liquid (PLDR
< 0.11), mix (0.11 < PLDR <0.15) and ice (PLDR>0.15), respectively. As expected, the percentages
determined by this study in Table 4b are similar to the results in Table 4a, the percentage of liquid
clouds classified by this study is ~10 to 20% lower, but ~10 to 15% higher in mixed-phased clouds
compared to those classified from both the methods of MLR and Shupe et al. (2005). The ice clouds
classified from this study are ~15% higher than those detected by MLR but ~10% lower than those
classified using Shupe. (2005) and Intrieri. (2002) method. This comparison is very reasonable given
that our method is developed from aircraft in measurements and radar-measurements over SO, while
the method developed by Intrieri et al., 2002 and Shupe et al., 2005 were based on the ground-based
lidar measurements over Arctic regions and MLR uses a machine learning algorithm trained over the
in-situ cloud and drizzle droplet measurements (CDP+2DS). The other reason for the difference lies in
the in-cloud constraints (LWC>0.001 g/m$^3$ to define in-cloud samples) used in our method which were
not used for the other two methods. Furthermore, MLR also reported a significantly high number of
unclassified cloud samples (~56%) for aircraft-measured in-situ temperatures above freezing point
(>0°C) which were not included in this phase-percentage calculation for low clouds using MLR (Table
4(b) column 2).



If we treat the results classified in this study as a reference, the lidar median PLDR values to classify
liquid, mixed and ice clouds may need to be tuned slightly for SO low clouds. The existing PLDR
thresholds (<0.11 for liquid, 0.11-0.15 for mixed, and >0.15 for ice phase clouds) as defined by Sassen
(1991), Intrieri (2002), and Shupe (2007), were originally established for Arctic clouds, which are
characteristically different from the MBL clouds over the Southern Ocean (SO). Using the classified
results in this study as a reference, we tune the existing HSRL PLDR thresholds for SO low-level clouds
and have the updated thresholds of PLDR < 0.09 for liquid phase, 0.09-0.18 for mixed phase, and >0.18
for ice phase clouds. This adjustment was based on a simple analysis of the low cloud samples measured
simultaneously by both radar and lidar. Further scrutiny may be necessary to estimate the accuracy of
these thresholds for low-level clouds over SO, and this could be a focus for future research.
**Table (4a). Comparison of the phase determination between MLR method cloud phase product**
**(D'Alessandro et al., 2022) and this study matched at the same temporal resolution (10 secs).**
**Presented number are raw sample counts.**

| MLR Method/ This Study | Ice (this study) | Mix Phase (this study) | Liquid (this study) |
|---|---|---|---|
| **Ice (MLR)** | **162** | 16 | 8 |
| **Mixed Phase (MLR)** | 67 | **140** | 64 |
| **Liquid (MLR)** | 426 | 457 | **995** |


**Table (4b). The cloud phase partitioning for each phase-type determined using this method**
**(dominant phase) compared with the MLR method and Shupe (2005), Intrieri (2002) method for**
**the classified low cloud samples during SOCRATES. The data is aggregated to a 10 second sample**
**interval. The unclassified cloud samples in the MLR cloud phase product are not included in the**
**sample % calculation in column 2, and the in-cloud constraint (LWC>0.001 g/m$^3$) is not included**
**for phase detected by MLR (column 2) and Shupe (2005), Intrieri (2002) (column 3).**

| | This Study | MLR Method | Shupe et al., 2005; Intrieri et al., 2002 Method |
|---|---|---|---|
| **Liquid %** | 45.4 | 71.7 | 52.3 |
| **Mix %** | 22.2 | 10.3 | 5.5 |
| **Ice %** | 32.5 | 18.0 | 42.2 |

As previously mentioned, it's important to note that these three classification methods are different.
The MLR method determines cloud phase based on tuning a MLR (multinomial logistic regression)
model to cloud hydrometeors sampled using the in-situ probes (CDP+2DS) onboard the NCAR/GV
aircraft during SOCRATES, while we used both in-situ and radar measurements in this study. The
HSRL lidar method is purely dependent on the PLDR thresholds. The HSRL lidar detects a smaller
fraction of the cloud fraction compared to the HCR radar, as lidar is highly attenuated for thicker cloud
layers whereas HCR radar can offer a well-resolved cloud profile. Consequently, the radar and lidar do
not provide measurements for the exact same cloud layers, with an overlap region of only about 8%.
Therefore, while the comparison between these three methods is not entirely straightforward, it provides
a reasonable rough estimation for comparing the phase estimations across a linear time dimension.






### 4.3 Cloud characteristics for each determined cloud phase

Table 5 lists the summarized macrophysical cloud properties for each classified phase, based on the 1-dimensional dominant phase from all 15 research flights during SOCRATES. The statistical results listed in Table 5 include sample counts (and percentages) along with mean and standard deviation for cloud-base and -top temperatures ($T_{base}$ and $T_{top}$) and heights ($H_{base}$ and $H_{top}$), cloud thickness ($\Delta H$) and LWP.

**Table 5. Summaries of cloud macrophysical properties for each determined cloud phase**

| Phase | Samples | $T_{base}$ (ºC) | $T_{top}$ (ºC) | $H_{base}$ (km) | $H_{top}$ (km) | $\Delta H$ (km) | LWP (g/m$^2$) |
|---|---|---|---|---|---|---|---|
| **Ice** | 1043 (~32.5%) | -3.5±4.6 | -6.0±4.8 | 1.12±0.63 | 1.59±0.61 | 0.47 | 2.7±2.6 |
| **Mixed-phase** | 712 (~22.2%) | -2.1±4.3 | -8.2±4.8 | 0.74±0.53 | 1.65±0.60 | 0.91 | 200.5±267 |
| **Liquid** | 1458 (~45.4%) | -1.2±4.9 | -3.9±5.4 | 0.90±0.54 | 1.34±0.59 | 0.44 | 89.7±100 |

### 4.3.1 Cloud-base and -top temperatures ($T_{base}$ and $T_{top}$)

Low clouds generally exhibit higher temperature trends than the recorded aircraft temperature at the actual flying levels because of the difference in tropospheric altitude of the flight and the actual cloud boundaries. The ERA5 air temperature is used to extract the temperature at the cloud base and cloud height altitudes. Figure 8(a) shows the occurrence probabilities of the estimated cloud phases against the ERA5 air temperature for the entire cloud transect, highlighting that all the cloud phases have the highest occurrence in the range of -5 to -2.5 ºC, while no ice and mixed phase exist at temperatures greater than 0 ºC, 100% liquid-phase concentration is observed at T >0 ºC. The frequency distributions in Fig. 8 (b-c) show that all $T_{top}$ samples from three phases increase monotonically from -20 ºC, peak at -7.5 and -5 ºC for mix and ice respectively, and -2.5 ºC for liquid-phase, and quickly vanish after 0 ºC except for liquid samples. The frequency distributions of $T_{base}$ samples from three phases almost mimic their $T_{top}$ counterparts but with different peaks: The maximum frequency of liquid and ice phase occurs at -2.5 ºC, while mixed-phase $T_{base}$ remains at 0 ºC. The different peaks in $T_{base}$ samples from three phases have reflected in their mean $H_{base}$ where the mean ice-phase $H_{base}$ is 1.12 km, higher than other two $H_{base}$ (0.74 and 0.90 km). $T_{base}$ and $T_{top}$ for liquid phase have the highest frequency at near -1 ºC. Ice and mixed-phase cloud temperatures show similar trends with most samples around lower temperatures. Interestingly the peaked $T_{base}$ of mixed-phase clouds occurs at 0 ºC because most of the mixed phase cloud samples occur around the cloud base where their temperatures are higher than cloud-top ones. It should also be noted that this analysis considers only the dominant phase for each layer (Fig. 8b-c), but the 2-dimensional phase is exclusively liquid for temperatures greater than 0 ºC (Fig. 8a) as discussed in the phase determination method.

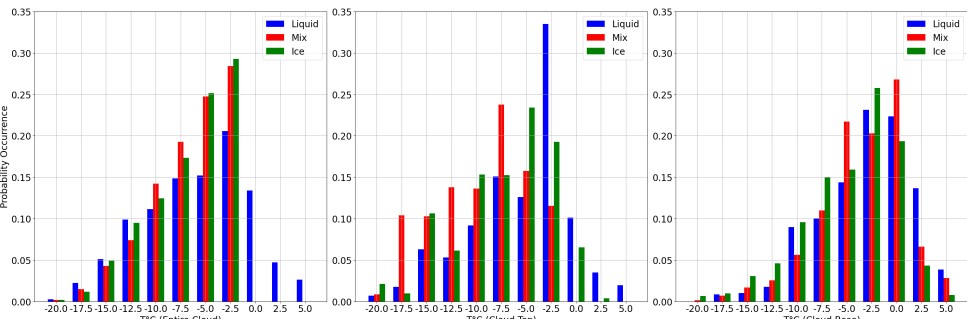

**Figure 8. Probability distribution of the entire cloud layer temperature (T) from ERA5 air temperature, cloud-top ($T_{top}$) and cloud-base ($T_{base}$) temperatures for each determined phase**

### 4.3.2 Profiles of determined cloud phase and radar observations

Figure 9 shows the vertical distributions of classified liquid, mixed-phase and ice cloud samples, as well as the total samples of LOW clouds from 0 to 3 km (retrieved from the 2-dimensional cloud phase profile). As mentioned above, liquid clouds are dominant, and its occurrence has the highest frequency around 0.75 – 1.2 km. The ice cloud occurrence follows the trend of liquid clouds with the higher frequencies at the levels of 0.75-1.5 km. Differing to liquid and ice clouds, the mixed-phase occurrence is evenly distributed in the cloud layer with higher sample counts from 0.5 km to 1.5 km. It should be noted that the sample counts of all three phases diminish to 0 at around 150 m which is where the estimated cloud base lies for low clouds in this study.

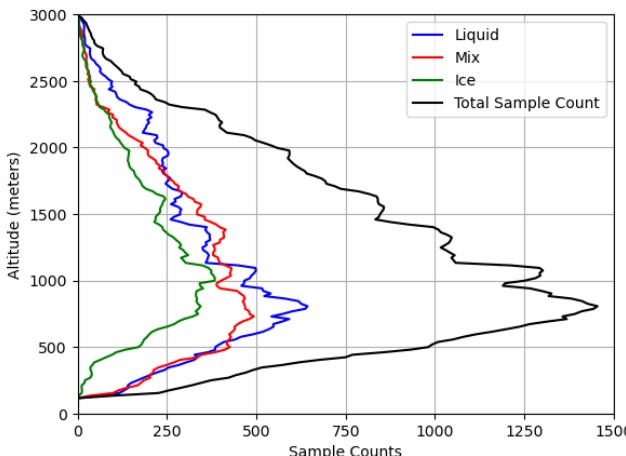

**Figure 9. Profiles of the cloud samples for each determined cloud phase along with the total sample number (black line).**

To further investigate the vertical distribution of classified liquid, mixed-phase, and ice clouds in LOW clouds during SOCRATES, we plot the normalized vertical distributions of HCR reflectivity (dBZ), Doppler Velocity (m/s) and Spectrum Width (m/s) in Fig. 10 (Contoured Frequency Altitude Diagram, CFAD).

Figures 10a-10c show the CFADs of determined liquid cloud samples where most of radar reflectivity dBZ values range from -35 to -25 with the median values of ~-25 dBZ, except for lower bottom regions of the cloud (normalized height, $H_i$ < 0.2).. The nearly constant median values with


height, and moderate dBZ, $V_d$ and WID, indicate the liquid cloud microphysical properties vary slightly
within cloud layer with a moderate range of cloud droplets. The maximum occurrences of $V_d$ and WID
for liquid clouds are ~ 0.0 m/s and 0.2 m/s. Much higher $V_d$ and WID at the low bottom regions indicate
that there are some large drizzle drops with a broader size distribution, however the number
concentrations of these large drizzle drops are not higher enough to significantly contribute radar
reflectivity. Based on the aircraft in situ measurements during SOCRATES (Zheng et al. 2024), the
cloud droplet and drizzle radii near cloud base are one order of magnitude difference (7 µm vs. 70 µm),
whereas their number concentrations are four orders of magnitude difference (100 $cm^{-3}$ vs. $10^{-1}$ $cm^{-3}$).
These lower number concentrations may attribute to the lower reflectivity near cloud base in Fig. 10a.
Another possible reason to explain the contradicted relationship between radar reflectivity and $V_d$/WID
is not enough liquid samples at the lower bottom regions, which can't be ruled out.

Compared to the CFADs of liquid cloud samples, the mixed-phase clouds have a broader and higher
radar reflectivity dBZ values with the maximum frequencies occur at ~ -10 dBZ around mid-cloud layer,
presumably due to their larger particle size and irregular shape or morphology. Correspondingly, their
median values are also much large (-15 ~ -10 dBZ), except for the lower bottom regions ($H_i$ < 0.2). The
CFADs of $V_d$ for mixed-phase clouds mimic the shape of liquid cloud samples but with higher
maximum occurrence at ~0.5 m/s and their median values increase monotonically from cloud top (~
0.6 m/s) to cloud base (~1.3 m/s), indicating that cloud droplets or ice particles increase from cloud top
to cloud base, much faster at the lower bottom regions. Consequently, there are more large drizzle drops
or ice particles near cloud base with significant downwelling movement and the prevalence of a broader
size distribution of particles. Surprisingly, the CFADs of WID for mixed-phase clouds are similar to
those of liquid clouds, but not as broader as liquid clouds at the lower bottom regions. These results
suggest that the well-mixed cloud droplets/drizzles and ice particles at the lower bottom regions make
particle size broader and larger WID values, but they are not as broader and higher WID as liquid clouds
where more both cloud droplets and drizzle drops co-exist near cloud base.

For ice clouds, their CFADs of $V_d$ are similar to those of liquid clouds, however, their CFADs of
radar reflectivity and WID significantly differ to those of liquid and mixed-phase clouds. Most of radar
reflectivity dBZ values range from -45 to -25 at the upper regions of cloud layer with the median values
of ~-30 dBZ. Surprisingly, these ice particles with lower radar dBZ values have a much higher WID
value, up to 0.8 m/s, and lower $V_d$ values (~ 0-0.2 m/s). These results indicate that most ice particles
for SO low clouds have small particles but with a broader size distribution. The CFADs of ice clouds
at the upper regions of cloud layer are consistent to those CFADs of liquid and mixed-phase clouds at
the lower bottom regions, suggesting that the lower dBZ regions where small cloud droplets or ice
particles are dominant can have a broader size distribution for SO low clouds.



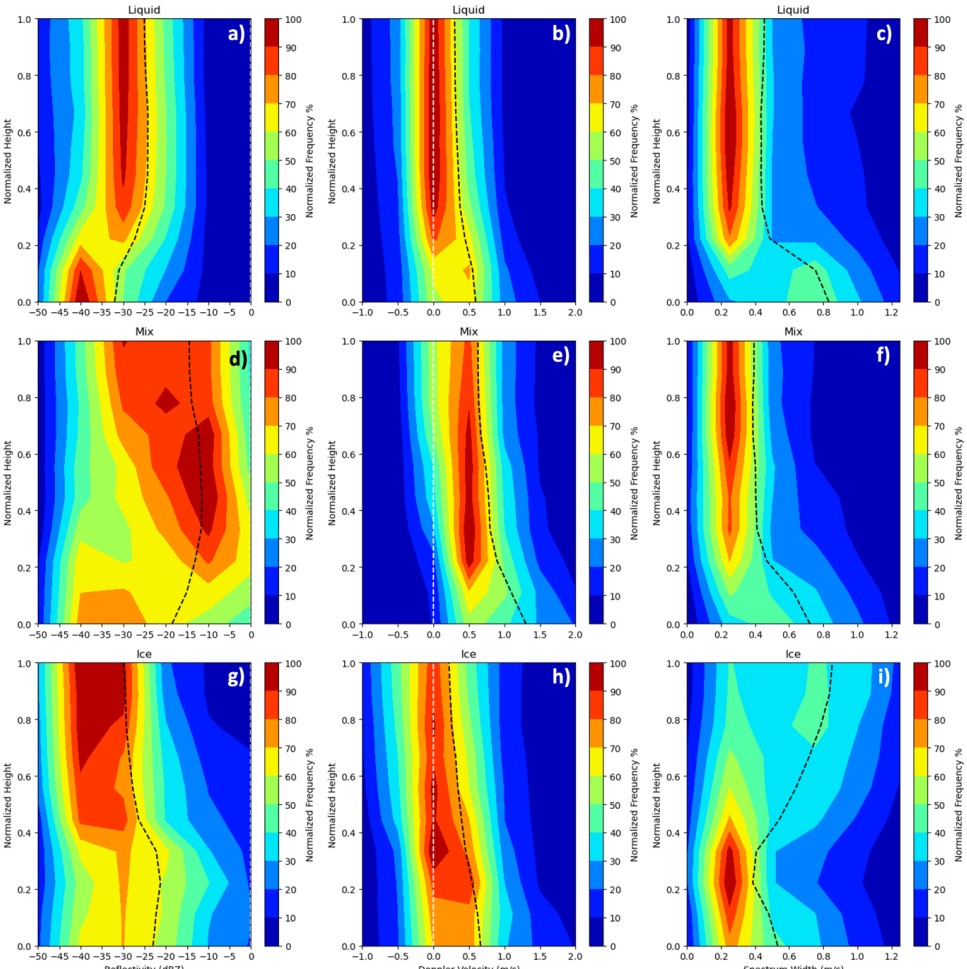


**Figure 10. Normalized vertical distributions of radar reflectivity (dBZ), Doppler velocity $V_d$ (m/s)**
**and spectrum width (m/s) for the classified liquid (a-b), mixed (d-f), and ice (g-i) clouds. during**
**SOCRATES. Height normalization is determined by $H_i = \frac{H - H_{base}}{H_{top} - H_{base}}$, where the cloud top is**
**denoted as 1 and base as 0. The median values are represented using the black dashed lines and**
**the white lines in $V_d$ denote 0 m/s. The colorbar denotes the occurrence frequency (%).**

$V_d$ and WID are codependent variables as $V_d$ indicates the motion of the hydrometeor samples
moving away or towards the radar, however WID is indicative of the spread or variability in the velocity
distribution. For an example, a positive (negative) $V_d$ represent a downdraft (updraft) motion, but a high
(low) WID indicates a significant variability in the velocities, including greater (lower) turbulence and
wind shear, which can be interpreted as a broader (narrower) size distribution. The regions with high
WID but low $V_d$ can be explained as that the average velocity of the cloud particles within the radar's
sampling volume is low, but with a significant spread or variability in their individual velocities due to
significant turbulence, wind shear, or different particles sizes. Conversely, a high $V_d$ but a small WID
suggests the either large drizzle drops, or ice particles are moving (downward) rapidly but uniformly
with a narrow size distribution.





The CFAD plots in Fig. 10(a-i) display noticeable skewness. Generally, the frequency plots are
skewed to the right, but in Fig. 10(d), the dBZ frequency for mixed-phase samples is skewed to the left
due to the higher reflectivity values observed for mixed-phase clouds. Moreover, the median values
(black dashed lines in Fig. 10) are significantly shifted towards higher values of WID and Vd near the
cloud base, while they shift towards lower values in dBZ for liquid and mixed phase clouds in Fig. 10(a
and d). This phenomenon can be attributed to the majority of cloud droplets exhibiting lower reflectivity
yet significantly higher values for spectrum width and Doppler velocity at the estimated cloud base.
Contributing factors include lower sample counts, and increased turbulence or wind shear contributing
to evaporation of particles around the estimated cloud base. Additionally, this observation also results
from the nature of cloud sampling during SOCRATES as demonstrated in Fig. 7(a-c), which introduces
greater volatility in recorded datasets due to the proximity of the aircraft to the cloud layers compared
to ground-based or satellite remote sensing.

**5 Summary and Conclusions**

This study developed a phase detection method using HCR radar and in-situ measurements to determine
the cloud phase over the Southern Ocean for low-level clouds sampled during the 15 research flights of
the SOCRATES campaign. The macrophysical properties and statistical results for the different types
of clouds and their correspondingly classified phases were discussed. Finally, the vertical distribution
for each phase-specific radar retrieved parameters was presented. Comparisons of this study with
existing literature were also discussed. The following conclusions were finally drawn:
1.  A new method based on radar reflectivity and spectrum width gradient was developed to estimate
cloud boundaries and classify cloud types as LOW, MID, and MOL based on the estimated cloud-
top and -base heights. LOW-type clouds with $H_{base}$ and $H_{top}$ below 3km were found to be the most
prevalent with almost 90% occurrence frequency. Liquid Water Path was calculated for each cloud
type using the estimated cloud heights and a merged CDP+2DS LWC measurement, with an
uncertainty of around 10 g/m$^2$. Average LWP values for LOW, MOL and MID clouds are 96.7 g/m$^2$,
109.48 g/m$^2$, and 27.9 g/m$^2$, respectively.

2.  A phase determination method was developed to classify the single-layered low-level (LOW)
clouds as liquid, mixed, and ice phases with the occurrence frequencies of 45.4%, 22.2% and 32.5%,
respectively. Comparison with the MLR phase detection method by D'Alessandro et al. (2021), and
Shupe et al. (2005) and Intrieri et al. (2002) method which used lidar PLDR thresholds, showed
that the percentage of liquid clouds classified by this study is ~10 to 20% lower, but ~10 to 15%
higher in mixed-phased than the results from other two methods, while the classified ice clouds
from this study are ~15% higher than those detected by MLR but ~10% lower than those classified
using Shupe. This comparison is quite reasonable, as our method is derived from aircraft
measurements and radar observations over SO while the methods developed by Intrieri et al. (2002)
and Shupe et al. (2005) were based on ground-based lidar measurements in Arctic regions.
Additionally, the MLR method employs a machine learning algorithm trained on in-situ cloud and
drizzle droplet measurements (CDP+2DS). Using the classified results in this study as a reference,
we tune the existing HSRL PLDR thresholds for SO low-level clouds and have the updated
thresholds of PLDR < 0.09 for liquid phase, 0.09-0.18 for mixed phase, and >0.18 for ice phase
clouds.

3.  For the low-level clouds from 0 to 3 km, the mixed-phase cloud dominates near cloud base (<1 km)
but are well distributed along the vertical cloud layer which could be attributed to large drizzle
drops or ice particles. The ice-phase clouds are prevalent from the mid to top cloud level (1-3km),
while most of the liquid-phase clouds are located in the lower mid-cloud range (from 500 m to 1
654        km).



4.  The normalized vertical profiles (CFADs) of radar reflectivity, Doppler velocity (Vd) and spectrum (WID) for each determined cloud phase show that the liquid and ice clouds have the lowest reflectivity values, with median reflectivities of around -30 to -25 dBZ, while mixed-phase clouds have a higher median reflectivity of around -15 to -10 dBZ due to large drizzle drops or ice particles. Higher Doppler velocity and Spectrum Width at the cloud base indicate greater drizzle or particle concentrations with significant downwelling movement and the prevalence of a wider size distribution of particles. The CFADs of ice clouds at the upper regions of cloud layer, lower dBZ values but larger WID values, are consistent to the CFADs of liquid and mixed-phase clouds at the lower bottom regions. These results indicate that small cloud droplets or ice particles (lower dBZ) for SO LOW clouds can have a broader size distribution (large WID).

In conclusion, the results presented in this study provide comprehensive statistical and phase-relevant macrophysical properties for the low-level clouds sampled during the SOCRATES campaign, along with presenting new methods to estimate cloud boundaries and determine the dominant cloud phase. These results would improve the current understanding of the low-level Southern Ocean cloud properties and further aid in improving model simulations and better representation of the climate.

**Data Availability.** All the data and the relevant instrumentation details (radar-lidar and in-situ) for the NSF SOCRATES campaign used in this study are freely accessible from the EOL data archive https://data.eol.ucar.edu/dataset/ and the official website https://www.eol.ucar.edu/field_projects/socrates. The MLR cloud phase product dataset is available at https://doi.org/10.26023/S6WS-G5QE-H113.

**Author contributions**. The idea of this study was discussed by AD, BX, and XD. AD performed the analyses and wrote the paper. AD, BX, XD, and XZ participated in the scientific discussions and provided substantial comments and edits on the paper.

**Competing interests**. The contact author has declared that neither they nor their co-authors have any competing interests.

**Acknowledgments.** The aircraft measured dataset and the SOCRATES campaign relevant details along with the list of publications are available free to access at https://www.eol.ucar.edu/field_projects/socrates. Special thanks to John D'Alessandro (University of Washington) for helping understand the working of the MLR method for phase determination, Ulrike Romatschke (NCAR) for explaining the radar-lidar constrained fuzzy logic parameters for the cloud and drizzle hydrometeor types, and Christopher J. Webster (NCAR) for helping to troubleshoot and run the XPMS2D software to visualize the 2DS probe imagery. The draft of this paper was briefly proofread using the AI tools Grammarly and ChatGPT for final revisions.

**Financial Support.** This work was supported by the University of Arizona's IT4IR TRIF and Provost Investigation funds. The researchers at the University of Arizona were also supported by NSF grant AGS-2031750 at the University of Arizona.

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
