# Peer review of "Cloud phase estimation and macrophysical properties of low-level clouds using in-situ and radar"

_Atmospheric Measurement Techniques, 2024_

## Author Comment (AC1)

**Response to Reviewer:**

We thank and appreciate the reviewers for their valuable scrutiny and feedback on our manuscript and the opportunity to submit a revised version. All the suggested changes and feedback have been duly addressed and implemented, with our responses for each comment presented as follows.

The major changes are:

1. A supplementary material has been added alongside to the revised manuscript containing the validation of the estimated cloud base and LWP calculations.

2. The CFAD analysis is completely revised.

3. The wording of the manuscript has been significantly improved to remove redundancies and increase clarity. Additions have been made as per reviewer suggestions.

Note: line numbers must have changed in the revised manuscript.

**Reviewer 1 Comments:**

**Question 1:** The authors developed a radar-based methodology to derive cloud base. However, they themselves state that it does not agree with the standard method based on lidar observations. The lidar-based method has been proven to work well in many previous studies and since the authors do not independently validate their new method the reader concludes that the radar-based method is not accurate.

**Response:** While it is true that lidar-based methods have been established for cloud phase identification primarily by Sassen (1991), Intrieri (2002), Shupe (2007) and been used in many literatures, however all those studies focused mainly on Arctic mixed-phase Clouds which have different characteristics compared to those over the Southern Ocean (SO) studied in this paper. Furthermore, using solely lidar-based evaluation is not accurate because lidar signals attenuate easily for optically thick cloud layers (Sassen, 1991), particularly for aircraft flew on the top of cloud layer (>80%). Hence, most studies over the SO either uses a synergistic method based on a combination of in-situ measurements and radar-lidar remote sensors onboard as shown by D'Alessandro (2021), Romatschke and Vivekanandan (2023).

Our analysis showed that while HSRL lidar backscatter signal was efficient for estimating cloud base but only when the flight flew below the cloud base offering a zenith-viewing direction for the lidar, which is also presented in Kang (2024). When aircraft flew on the top of clouds, it is impossible to use lidar signals to estimate cloud bases. However, the HCR radar reflectivity and spectrum width profiles were found to be comprehensive enough for estimating both cloud top and cloud base heights at both the zenith and nadir-pointing directions. Similarly cloud base estimation using lidar at the nadir pointed view was found to be erroneous and we estimated an offset of ~400 meters between the cloud base estimated by the two instruments (HCR and GV-HSRL).

Hence, the HSRL measurements are not optimal for phase-estimation of SO clouds sampled during the SOCRATES campaign, while HCR measurements are more robust. The significance of millimeter-wavelength radar for cloud observation has been widely acknowledged (Huang, 2019; Hobbs, 1985; Lhermitte, 1987; Kollias, 2005, etc.). The objective of this paper is to develop (propose) an airborne radar-based phase estimation method which will comprehensively estimate low-cloud phase since low-clouds over the SO sampled during the SOCRATES campaign which have exhibited to have thicker cloud layers (~3 km) which the lidar signals cannot detect accurately. The sampled low-level clouds also display significantly lower concentration of larger ice particles ($>10^{2.5}$ µm) as shown in Figure 2, which is where the lidar signals might have been a more useful addendum to the phase-estimation algorithm. The reliability of a radar-based phase estimation method for SO clouds has already been established in Xi et all (2022) using data from the ship-based MARCUS campaign. The validation of the results has been carried out by comparing it to some existing published methodologies for cloud phase-estimation, like D'Alessandro (2021), Shupe (2005, 2007), and Intrieri (2002), on the same SOCRATES low-cloud dataset, which has yielded close agreements in phase percentage retrievals for Liquid, Mixed and Ice phases. Finally, the usage of aircraft-based remote sampling data makes this phase-estimation method closest to the ground truth we can have currently due to the proximity to which the aircraft can fly near the cloud layers.

**2:** The authors then develop a new cloud phase classification method based on in-situ and radar observations. The method heavily relies on spectrum width observations but fails to account for shortcomings in the spectrum width observations, which leads to inaccurate conclusions.

**Response:** Using radar-measured spectrum width as one of the key parameters for cloud phase-estimation is not unprecedented and its reliability as a phase-identification factor has been presented in multiple if not all existing phase-evaluation methodology studies including Xi (2022), Shupe (2007), Romatschke and Vivekanandan (2023), Desai (2023) across multiple campaigns as well. The simple logic behind this being lower spectrum widths are exhibited by homogenous single-phase cloud particles (ice and liquid) while higher values are exhibited by mixed-phase clouds. However, it should be noted that spectrum width is used along with doppler velocity and reflectivity to correctly estimate phases, where a combination of all three measurements is used together and not in separate steps. This has been explained thoroughly in Section 4.1 and Figure 6. The synergistic usage of all three remote-sensed measurements i.e. reflectivity, spectrum width and doppler velocity has been done considering the methodology in Shupe (2007), although we developed the thresholds based on the nature of the SOCRATES-sampled datasets specific to the low-clouds over SO for this study. The usage of aircraft-sampled dataset for the radar reflectivity, doppler velocity and spectrum width make this closest to the ground truth we can achieve over the SO, due to the proximity of the aircraft to the cloud layers compared to ground-based or satellite remote sensing, which also adds a certain degree of volatility in retrievals which has been mentioned in Section 4.3.2.

**3:** It also relies on the liquid water path, which is derived along the aircraft track from in-situ measurements and then applied to the whole vertical column of the radar observations. In my opinion, it is not an accurate assumption to claim that LWP is constant throughout a cloud. The authors themselves state that the resulting 2D classification does not look accurate. Nonetheless, they collapse it back into a 1D classification and use both throughout the rest of the study.

**Response:** The validation of the LWP calculation has been carried out and added to the supplementary. In short: Validating the calculated LWP as in eq. 3 in the manuscript with the LWP measured across the sawtooth regions where the sampling height is the flight altitude and LWC is measured along that same height as well (i.e. LWP = LWC*flight_altitude for the sawtooth regions) shows excellent agreement. The correlation value (r) is found to be 0.85 with a strong statistical significance (p<<0.001). Further comparing and validate LWP calculations with existing geographically and spatio-temporally collocated measurements during the MARCUS campaign which provided radiometer measured LWP values over the SO. We compare the LWP by creating a lat-lon grid box with overlapping measurements for low level clouds between the two campaigns SOCRATES and MARCUS for the similar climatological months (Jan, Feb and March), and found excellent agreement in the resulting values. LWP measurements also show an excellent agreement with mean values: 110 g/m$^2$ (SOCRATES, as calculated in this study) and 127 g/m$^2$ (MARCUS, radiometer-measured), and a difference of ~17 g/m$^2$.

Currently there is not much available methods known to us to reliably estimate LWP as a height-resolved variable from in-situ measured LWC, furthermore, LWP is not a directly measured variable during the SOCRATES flight-tracks. Frisch (1995, 2002) does offer a way to estimate vertical profiles of LWC and R$_e$ (effective radius) as a function of radar reflectivity Ze and number concentration, but it needs to be majorly corrected for signal attenuation, which is beyond the current scope of this paper. The aircraft measures cloud particle and drizzle size distributions along the flight-track which has been correctly identified by the reviewer, but also -above, -below and inside the cloud layers during the whole sampling leg. We used this in-situ measured PSD and DSD to calculate cloud and drizzle LWC following the method in Kang (2021) and Zheng (2024), and then estimate the LWP. Since our paper is not focused on finding novel LWP-calculation methods, we relied on the simplistic method of integrating LWC measurement across the entire cloud vertical profile following Oh (2018) and Mioche (2017), the calculated LWP as a function of cloud thickness might have some offset from a radiometer measured LWP but is not inaccurate when calculated over multiple range gates (of 19.2 m each). At 150 m/s (avg. cruising speed) * 10 secs (temporal averaging) ≈ 1.5 km, clouds are reported to be globally homogenous This method based on single-frequency radar retrieval does have the limitation of assuming of weak vertical changes in microphysical properties within 500 m thick cloud layers (Tridon, 2013b), and can be mitigated through an averaging of radar observations (Chandra, 2015) as has been done in this study. The LWC and LWP uncertainty has been taken care of and duly mentioned in the literature. We have acknowledged that this is a coarser LWP estimation but this method of finding LWP from LWC as a function of cloud thickness has been in standard use in the industry for very long, including Oh (2018), Mioche (2017), Toledo (2021). Furthermore, it should also be noted that LWP is used along with the remote-sensed parameters for phase-estimation and not uniquely. The estimated LWP was also compared with reported LWP measurements over the SO for low-clouds, and we found a decent agreement in the values (Xi, 2022; Mace, 2023; Tan, 2023).

We did identify a typographical error or lack of clarity in our writing in Section 4.2 which might have given the reviewer a feeling that we are not confident in the 2D phase retrieval, what we wanted to say was that the 2D phase profile presented in Figure 7g is just a small transect of the entire cloud profile, chosen for offering the best visual clarity, and is not depictive of the actual 'dominant' cloud phase (Fig 7h) on which the second half of this paper is based on i.e. validating the findings based on the statistical analyses. We carry out a scrutinized estimation of the dominant phase based on sample counts for each phase along each vertical layer (based on the temporal interval) which has been explained in the revised manuscript. We have

improved the writing for this part to ensure no further confusions are created due to the nature of wording or typographical errors.

**4:** The authors compare some of their results with methods from previous studies (even though for some reason they do not compare it with the phase classification that is provided in the radar/lidar dataset itself) and note that their results do not agree very well with previous studies.

**Response:** We have not identified any standard dataset in the SOCRATES repository that provides phase-identification directly from radar-lidar readings. All phase-related work is normally a modified product by independent research teams who have developed some method to identify cloud phase, and each is unique and independent based on the underlying methodologies developed like D'Alessandro (2021), Romatschke and Vivekanandan (2023). We tried to compare our phase estimation results with MLR algorithm-based phase identification by D'Alessandro (2021) and lidar PLDR-based estimation method developed for the Arctic clouds using ground based lidar observations by Shupe (2005, 2007) and Intrieri (2002). We have duly mentioned in the literature that these three classification methods are different yet as highlighted in Table 4b there is some decent level of agreement in the results based on the percentages for total liquid, mixed and ice phase distributions for the low-level SO clouds. The dominant phase retrieval from this study has an agreement of around 55.5% with results from MLR. The MLR method determines cloud hydrometeor phase based on tuning a MLR (multinomial logistic regression) model to cloud hydrometeors sampled using the in-situ probes onboard the aircraft during SOCRATES only at the flying or sampling altitude of the SOCRATES aircrafts, while we used both in-situ and radar measurements in this study. The lidar method is purely dependent on the PLDR thresholds. The lidar detects a smaller (spatial) sample of the cloud fraction compared to the HCR radar, as lidar signal is highly attenuated for thicker cloud layers (along the vertical) whereas radar can offer a well-resolved cloud profile. Therefore, while the comparison between these three methods is not entirely straightforward, it provides a reasonable rough estimation for comparing the phase estimations across a linear time dimension.

**5:** I am not sure why the authors chose to develop their own method, when proven methods are readily available, and then decide to use their own method, even though they themselves have found that it does not produce good results.

**Response:** We are aware of some of the existing published methods of phase identification, acknowledge the quality of those methodologies, and duly acknowledge them while developing our study. However, phase identification for low-level Southern Ocean (SO) clouds is a work in progress and significant more commitment and future studies are needed to develop data products and methodologies that can be used as industry standard. Every existing study has their own unique methodology using datasets from multiple platforms, focusing on multiple geographic regions of study, and tuned to specific cloud distributions over different climatic regions. Hence, each yield in different results. The lack of ample cloud phase-identification studies focused on low-level clouds over the SO drove us to develop our own method, using the SOCRATES aircraft campaign dataset which offers proximus sampling of the cloud layers and can be considered as closest to the ground-truth. Furthermore, using an array of in-situ and radar-retrieved parameters that do not attenuate easily for thicker cloud layers as observed over the SO. Low-level optically thick clouds are poorly misinterpreted in climate models, leading to greater uncertainties in estimating shortwave fluxes and predicting climate change. Understanding the dominant cloud phase and

phase-related statistics of the low-level SO clouds is crucial to expanding our current understanding of the region along with developing better parametrization for the increased accuracy of the global climate model predictions (Zhao, 2023; Liu, 2023). This clarification and the need for development of more phase-classification methodologies and data products, and the uniqueness of this study and their importance has been highlighted in the literature.

Furthermore, we note that the nature of wording we used in this paper might have confused the reader to believe that we are stating our results to be inaccurate, whereby we just wanted to acknowledge the levels of uncertainties or offsets from the methodology, volatility in measurements due to nature of sampling, and the biases arising from the selected cloud samples studied for this work. This was done with the best intention of doing free and fair scientific work and inviting critical analysis in the form of future work which would either further validate our methodology and findings or suggest ways to improve our current work. This is a piece of investigative research which presents our analysis and attempts to develop a new methodology aimed at addressing the common challenge of understanding cloud phase and macrophysical properties. We sought to employ the best possible means known to us, a challenge that has also been approached by other studies through various methods. The existence of previous studies and the acknowledgment of uncertainties or biases in the measurements and results should not be interpreted as a lack of confidence or inaccuracies in the methodology or findings, or to refute the significance of such work. This is standard industry practice in the pursuit of robust scientific work.

**Reviewer Specific Comments:**

**6.** Line 147: Please explain what is meant with "interpolated to fixed radar-heights at a range gate of 19.2 meters". The range gates are already fixed. And why 19.2 m?

**Response:** Yes, and that fixed range gate height for the HCR radar onboard SOCRATES is 19.2 m. There are negative range gate values for below surface level measurements, these were all corrected with respect to the relative height to the aircraft altitude, and to keep the range gates at a 19.2 m fixed height interval. We have rectified the writing.

**7.** Section 2.2: I am very skeptical about the method that derives cloud base. How was that method validated? Using data from the GV-HSRL to determine cloud base is the state-of-the-art methodology. If the method proposed by the authors does not agree with the GV-HSRL measurements, that would generally indicate that it is not accurate. If the authors really want to claim that their method is superior, they need to find an independent validation method. Simply plotting the results on the radar observations is not sufficient.

**Response:** We thank the reviewer for raising this question and acknowledge that this would improve the findings of our study. We do acknowledge that this is also a coarse approximation but is seen to work well in the absence of a ceilometer detected cloud height, as was the case with SOCRATES. Using the lidar estimated cloud base as in Kang (2024) resulted in huge offsets in cases when the flight flew above the layer and the radar/lidar pointed nadir because lidar signal attenuates for thicker cloud layers. For zenith pointing measurements, the lidar and radar determined cloud base height tend to overlap to a certain degree. We did not include this analysis in the manuscript or supplementary as it does not answer the proposed objectives of this research.

Similar to the LWP validation explained in response to comment 3, we also compare and validate our cloud base estimations with existing geographically and spatio-temporally collocated measurements during the MARCUS campaign which provided ceilometer and micropulse lidar estimated cloud base over the SO. We compare the cloud base height values by creating a lat-lon grid box with overlapping measurements for low level clouds between the two campaigns SOCRATES and MARCUS for the similar climatological months (Jan, Feb and March), and found excellent agreement in the resulting values. The mean cloud base height difference was found to be around 159 meters where SOCRATES mean cloud base height (derived using the method described in this study) was 1055 meters and for MARCUS was ~900 meters. This is expected as MARCUS being a ship campaign sampled more lower clouds (closer to the surface) than SOCRATES, and the mean difference is within a decent agreeable range for cloud base heights. Detailed explanation will be added to the Supplementary.

**8.** I would also like to point out that spectrum width in regions with low signal to noise ratios, which are often observed at the cloud edges, are often noisy and not all that reliable. Using their gradient to derive cloud boundaries is therefore not recommended. You can see in Figure 1c that there is a similar gradient at the top of the cloud as at the bottom. The argument that this gradient implies drizzle or precipitation (as stated at line 177) is therefore highly questionable. The fact that the algorithm finds cloud base in the middle of a cloud layer where there is clearly no precipitation at all (Fig. 1 around 26.44) is also suspicious.

**Response:** The gradient of Spectrum Width can be used to differentiate between the precipitation, drizzle and cloud layers as it is sensitive to particle sizes along with the spread of the doppler velocity spectrum. In Figure 1c, the presence of very high spectrum width values at the cloud base allowed us to presume the high incidence of drizzle and precipitation at the cloud base. The cloud base height retrieval method only considers the spectrum width gradient starting from the lowest radar height range gate and starts working towards the cloud top, one height bin at a time. When moving from the clear sky to the precipitation and/or drizzle layer at the cloud base, the gradient is the highest (or maximum positive), then where the large precipitation droplets and/or drizzle particle layer moves into the cloud layer with smaller cloud and drizzle particles we see the maximum negative change (or lowest gradient value for that vertical layer) in the spectrum width gradient, where we predict the cloud base to exist. Even if the spectrum width at around cloud top appears to be similar as at the cloud base for some instances, it will not have the same width gradient, otherwise the algorithm detects it. Also, the lowest height bin to have the maximum negative gradient is only considered to be the base height. This method works only on a bottom-up approach from the lowest height bin to the highest. As mentioned, the GV-HSRL retrieved cloud base is around 400 meters higher than using HCR on an average, which we found were incorrect especially in regions where the instruments pointed nadir.

Furthermore, Figures 1a-d are simply a small segment from RF01 which illustrates a small fraction of the dataset, hence the existing noise and outliers from the height estimation are prominent in this figure. For the entire dataset covering RF01 to RF15, the outliers and noise do exist as this is a coarse approximation, which is integral to a radar measurement but does not contribute majorly towards any misleading results as the final statistics are calculated based on analyzing the entire dataset together based on the central measurements (using median) which reduces the impact of outliers. Also between 26.42 and 26.44 it can be observed that there are valid cloud signals observed by both the reflectivity (Fig. 1b) and doppler velocity (Fig. 1d) measurements of the radar, hence spectrum width being either very low or invisible at that region could be attributed to stagnant cloud particles (as doppler velocity for that small

region is ~0 m/s, Fig. 1d), very thin non-precipitating and homogenous cloud layer (as is noticeable from the figures) or signal attenuation. Note: the time axis has been corrected so 26.42 is now 2.42 hours UTC (decimal time).

**9.** Table 2: What is meant with "may not be single layer"? Does this mean that they could be multi-layer or deep? I don't think it makes much sense to put both of these cloud types together in one category. The same is true for the HML category. That seems to be a deep cloud but the name of the category seems to indicate that it is a multi-layer cloud.

**Response:** The cloud classification (Table 2) is based on Xi (2010). For the categories that have been mentioned as may-not-be single layer are cases where the height classification holds but the reflectivity profile indicated one or more separate cloud layers present. Since the average flying altitude of SOCRATES aircrafts is around 6 km, we do not observe any significant instances of such double layered clouds. Furthermore, we do not use these category of clouds for any further analysis as this study depends solely on single-layered low level clouds (<3 km).

**10.** Line 217: What is meant with "due to the selected cloud cases"? What cases were selected? How many? And based on which criteria?

**Response:** It was an error in writing. Removed this sentence.

**11.** Section 2.4: I'm afraid I do not understand how LWP is calculated. Do the authors assume that the LWC is vertically constant throughout the cloud? That does not seem like valid assumption.

**Response:** See our response to comment 3, for LWP validation.

**12.** Figure 3: How do these results compare to similar classifications in the literature? (Mace and Protat 2018, Truong et al. 2020, Romatschke 2023)

**Response:** Currently it is not feasible or practical to carry out such a comparison as all the mentioned methods are very different and have different approaches and results. We did not identify any existing phase retrieval study that could offer a pixel-by-pixel comparison with this study. Hence an aggregate comparison is the best we could do here as is presented in Table 4a and 4b. We are however planning a future study that will comprehensively combine all the phase estimation studies including ours for a more robust and synergistic analysis of SO clouds.

**13.** Table 3: The LWP values have very large SDs. Are they actually meaningful?

Response: Yes this is expected and consistent with findings over the SO. The difference was insignificant when these results were compared to the average LWP values observed over the SO by previous studies like Xi et al. (2022) and Mace et al. (2021). The greater degree of uncertainty present in the estimated average LWP (Table 3) is consistent with other studies. Even for a greater precision calculation of constrained height-resolved LWC as in Huang (2009) and Hogan (2005) using dual-frequency radars, the retrieved LWP still showed a mean difference of 70-120 $g/m^2$ with microwave radiometer (MWR) measured LWP.

**14.** Section 4.1: The authors develop a new methodology to derive cloud phase from radar observations. However, they do not validate the method. This leads to the following questions: a) Why was a new method developed when a phase classification is already provided in the dataset? Why do the authors think their methodology is valid if it is not verified? Why do they believe that their method is better than previous methods (that have actually been validated)? Why was it not validated?

**Response:** Existing phase methodologies especially the UWILD characterization (Atlas, 2021) and/or the MLR method (D'Alessandro, 2021), Romatschke and Vivekanandan (2021) PID method, along with the lidar-based phase detection methods developed over the arctic like Shupe (2007) are valuable but characteristically quite different than our proposed methodology. A pixel-by-pixel is not possible with any existing datasets currently, and any such comparisons are not deemed apple-to-apple. However, the comparison as presented in table 4a and 4b show decent agreement between the overall findings of this study and existing methods and helps in validating our final results to some extent. Validation of our 'proposed' phase-retrieval method can only be carried out through further synergistic investigation comparing existing studies over the SO relevant to cloud-phase estimation. We are planning a future study on the same, but it is beyond the scope of the current manuscript.

**15.** At line 375 the authors say that they compared with previous work. What exactly does that mean? What were the results of these comparisons? Also, the classification heavily relies on LWP and spectrum width. As mentioned earlier, I am not sure how trustworthy LWP is, as it is assumed to be vertically constant (see first paragraph of Section 4.2). Again, spectrum width is noisy in low SNR regions and some, if not many, of the high values may be unreliable. (Romatschke and Vivekanandan decided not to use spectrum width in their algorithm for this reason.)

**Response:** The used radar signal (dBZ, WID and $V_d$) thresholds were compared with the thresholds defined in the referenced studies to check for consistency in phase-retrieval. Decent agreements were found in this step. Spectrum width has been consistently used for phase retrieval in existing studies like Xi (2022), and Shupe (2007). As per Romatschke and Vivekanandan (2021), the Doppler spectrum width field was also included in earlier versions of the PID algorithm but was later eliminated to simplify the algorithm. Radar signal does suffer from noise but those are majorly concentrated at near-surface levels where they face near-surface contamination problems related to echo.

**16.** In Section 4.2., the authors state that "phase retrieval method but may be not highly depictive of the actual cloud phase". In that case, what is the value of the 2D derived cloud phase? What is the purpose of this effort? The 2D cloud phase is then again collapsed into a 1D cloud phase product which apparently "returns reasonably accurate findings compared to other phase detection studies over the SO". These comparisons are later shown (see below) but they are not convincing.

**17.** What is the purpose of developing this new product if other phase classifications already exist? Why start with the 1D cloud probe observations, expand them to low-quality 2D observations, and then collapse them again into 1D phase estimations? Why not use the cloud probe measurements directly and the phase classifications that were derived from them in previous studies?

**18.** Starting line 442, the 1D phase classification is compared with phase classifications from other studies. Unfortunately, the classifications from the current paper do not agree well with the classifications from

previous studies. The authors speculate about the causes of these discrepancies but fail to explain why the reader should trust their results over those from previous studies.

**Response:** We believe we have provided sufficient justification for comments 16, 17 and 18 in Responses to comments 2, 4, 5, and 14.

**19.** Starting line 474 the authors suggest tuning thresholds from previous studies to match their results but I am not convinced that this is the right thing to do. Again, why do the authors think that their results are more accurate?

**Response:** Majorly the nature of the clouds as described in the Introduction section, which also accounts for the different microphysical nature, spatial heterogeneity, and the nature of ice-liquid phase partitioning in SO clouds compared to MBL clouds over the Arctic region. However, as observed, the PLDR thresholds we propose based on our findings in Section 4.2 are very close, if not at all the same as those defined over the Arctic. Further scrutiny will definitely help to better validate the results of our 'proposed' methodology.

**20.** Section 4.3: Given all the problems pointed out earlier (unproven cloud base, unreliable phase classification), I am not sure how trustworthy and valuable this analysis is.

**Response:** As mentioned, cloud base and LWP validation have been carried out and a detailed explanation will be added to the Supplementary. Furthermore, all the above responses can be referred to here to justify our findings and proposed further future investigations.

**21.** Section 4.3.1: For this analysis, is the 2D cloud phase or the 1D cloud phase product used?

**Response:** The analysis of $T_{top}$ and $T_{base}$ was based on the 1D dominant phase profile (Figures 4b, and 4c). Only Fig 4a was based on the 2D cloud phase as a pixel-by-pixel comparison was possible with the 2D ERA5 temperature.

**22.** Section 4.3.2: Given that the authors stated earlier that the 2D cloud phase product is not particularly accurate, why is it used here?

**Response:** The wording has been revised significantly in the revised manuscript to improve clarity for this part, especially relevant to the 2D phase profile. The 2D phase profile provides a way to visualize the vertical distribution of the phases along the cloud layers, hence is used here.

**23.** Starting line 551: I disagree with the author's interpretation of the CFADs. In my opinion, the CFADs for the liquid (and mixed) clouds show a strong bright band signature at 0.2 km normalized height. This bright band signature contradicts the author's claims that the clouds are predominantly liquid. This confirms that the phase classification algorithm does not produce high-quality results. Looking at Figure 7, one gets the impression that the algorithm tends to classify deep clouds with strong reflectivity as mixed, deep clouds with weak reflectivity as liquid, and thin clouds as ice. This is reflected in the CFAD categories of Figure 10. I do not see a connection with cloud phase. It is not surprising that the authors use the word "surprisingly" a lot in the description of the CFADs since the CFADs do not actually show clouds of the respective phases.

**Response:** We appreciate this comment as it helped us identify certain key issues in the CFAD presented, which led us to reanalyze the CFADs and the new CFAD is presented in the revised manuscript. The key changes are that since a direct pixel by pixel mapping of the phase profile was tampering the 2D time-height structure of the reflectivity, spectrum width and doppler velocity profiles. In the new CFAD we carefully remove all the surface values and only retain the profiles from the estimated cloud base height to the cloud top height eliminating near-surface contamination or noise. Furthermore, the time averaging is increased to 1 minutes from the previous 10 seconds to further smoothen the data to handle outliers and remove significant noise from the volatility of the remote sensed measurements. This helped significantly smoothen and polish the dataset prior to plotting the CFADs. The revised CFAD highlight the dependence of dBZ on particle size, while Vd and WID profiles reveal particle motion and turbulence, with findings showing turbulence-induced broadening near cloud bases and transitions in particle sizes and phases from cloud tops to bases. Mixed-phase and ice clouds show distinct patterns, with turbulence and particle growth playing critical roles in their vertical evolution. The new findings suggest a median dBZ of around -26 for liquid phase with a slight shift towards -30 from lower mid to cloud base (normalized height, $H_i<0.2$), dBZ for mixed-phase remains significantly high at around -20 dBZ for mixed phase displaying a greater spread which can be attributed to the wider variability in particle size distributions, ice phase displays a median dBZ of around -27 from cloud top to mid-cloud level but increases to -25 at the cloud base due to incidence of larger particle sizes as ice crystals grow from the cloud top to cloud base by accumulation and aggregation processes. The doppler velocity ($V_d$) shows similar trend across all the three phases increasing significantly from the mid to cloud base level ($H_i<0.2$) due to significant downwelling motion. The spectrum width (WID) of liquid and mixed phase cases are similar exhibiting sharp increase from $H_i<0.2$ indicative of significant downwelling motion and greater turbulence at the cloud base. For ice phase the median WID is significantly higher at the cloud top (0.7 m/s), decreases at mid cloud levels and again increases to ~0.6 m/s at cloud base which is indicative of a broader spread in the ice particle size distribution at the upper levels and significant turbulence and downwelling motion at the cloud base. The CFADs also confirm the irregular shape or morphology of ice particles along with the higher incidence of larger drizzle and ice particles at the cloud base, but not enough liquid samples at the lower bottom regions. The results from Maciel (2024) have shown the presence of stronger in-cloud turbulence and updraughts in the clouds sampled during SOCRATES, especially for cases when supercooled liquid droplets are surrounded by ice crystals and/or mixed phase cases. This is demonstrated by the nature of the CFADs where a stronger turbulence causes an increase in Vd and WID values near the cloud base ($H_i<0.2$). Detailed analysis is presented in the revised manuscript.

**24.** Line 586: The higher width values indicate that a significant portion of the data comes from low SNR cloud edges where width tends to be noisy.

**Response:** The new CFAD in the revised manuscript will help address this.

**25.** Paragraphs starting line 599 and 609: Same as previous comment. Please do not over-interpret the high width values in low SNR regions.

**Response:** The reviewer is correct in pointing out low SNR regions but the revised CFAD has tried to omit near-surface regions and minimize this error. The co-dependency of Vd and WID are however valid justifications for the interpreting the observed CFAD characteristics.

**26.** Conclusions: I am not convinced by the conclusions given all the issues pointed out in the previous comments.

**Response:** See response to previous comments.

**Technical Corrections**

**27.** I would like to point out that there are many(!) grammatical issues throughout the paper which are too numerous to list. Sentences are often very long and not only the reader, but apparently also the authors get lost. Two of many examples are the sentence starting line 33 and the sentence starting line 80.

**Response:** Thanks for pointing this out, the writing has been improved significantly in the revised manuscript.

**28.** Line 25: Please define "SLW".

**Response:** Corrected as suggested

**29.** Line 109 and throughout the manuscript: The instrument is called the "GV-HSRL".

**A:** Corrected it to GV-HSRL, but we keep just HSRL for the rest of the manuscript as it is indicative that we refer to the same High Spectral Resolution Lidar instrument. Helps in reducing redundancy and word-count.

**30.** Line 128: The SOCRATES acronym has already been defined earlier.

**A:** Corrected as suggested.

**31.** Line 132: Type RV -> GV.

**A:** Corrected as suggested.

**32.** Figure 1: Please correct the values on the time axis. (As far as I know, a day has only 24 hours.)

**Response:** The time axis has been corrected in the revised manuscript, and converted to have the corrected UTC decimal time.

**33.** Line 340: This sentence is a duplicate of the previous one.

**Response:** Corrected.

**34.** Line 554: When using CFADs, please cite Yuter and Houze, 1994.

Response: Cited suitable in revised manuscript. Thanks for this suggestion.

**35.** Figure 10: Altitude unit is missing.

**Response:** This is the normalized height as explained in the caption (Fig. 10), which is a ratio and hence do not have altitude units. 1 refers to cloud top height, 0 base.

**36.** Data availability: The authors only post a general link to the EOL Field Data Archive. It is important to list the individual datasets with their respective DOIs in this section.

**Response:** Corrected in revised manuscript.

**REFERENCES:**

Atlas, R., Mohrmann, J., Finlon, J., Lu, J., Hsiao, I., Wood, R., & Diao, M.: The University of Washington Ice-Liquid Discriminator (UWILD) improves single-particle phase classifications of hydrometeors within Southern Ocean clouds using machine learning, Atmospheric Measurement Techniques, 14(11), 7079–7101, https://doi.org/10.5194/amt-14-7079-2021, 2021.

Bazantay, C., Jourdan, O., Mioche, G., Uitz, J., Dziduch, A., Delanoë, J., Cazenave, Q., Sauzède, R., Protat, A., and Sellegri, K.: Relating Ocean Biogeochemistry and Low-Level Cloud Properties Over the Southern Oceans, Geophys. Res. Lett., 51, e2024GL108309, https://doi.org/10.1029/2024GL108309, 2024.

Chandra, A., Zhang, C., Kollias, P., Matrosov, S., and Szyrmer, W.: Automated rain rate estimates using the Ka-band ARM zenith radar (KAZR), Atmos. Meas. Tech., 8, 3685–3699, https://doi.org/10.5194/amt-8-3685-2015, 2015.

D'Alessandro, J. J., Diao, M., Wu, C., Liu, X., Jensen, J. B., & Stephens, B. B.: Cloud phase and relative humidity distributions over the Southern Ocean in austral summer based on in situ observations and CAM5 simulations, Journal of Climate, 32(10), 2781–2805, https://doi.org/10.1175/JCLI-D-18-0232.1, 2019.

D'Alessandro, J. J., McFarquhar, G. M., Wu, W., Stith, J. L., Jensen, J. B., & Rauber, R. M.: Characterizing the Occurrence and Spatial Heterogeneity of Liquid, Ice, and Mixed Phase Low-Level Clouds Over the Southern Ocean Using in Situ Observations Acquired During SOCRATES, Journal of Geophysical Research: Atmospheres, 126(11), https://doi.org/10.1029/2020JD034482, 2021.

D'Alessandro, J., Schima, J., McFarquhar, G.: SOCRATES Cloud Phase Product. Version 1.0. UCAR/NCAR - Earth Observing Laboratory [data set], https://doi.org/10.26023/S6WS-G5QE-H113, 2022.

Desai, N., Diao, M., Shi, Y., Liu, X., & Silber, I.: Ship-Based Observations and Climate Model Simulations of Cloud Phase Over the Southern Ocean, Journal of Geophysical Research: Atmospheres, 128(11), https://doi.org/10.1029/2023jd038581, 2023.

Dong, X., Das, A., Xi, B., Zheng, X., Behrangi, A., Marcovecchio, A. R., and Girone, D. J.: Quantifying the differences in Southern Ocean clouds observed by radar and lidar from three platforms, *Geophys. Res. Lett.*, [in review], 2024.

Frisch, A. S., Fairall, C. W., and Snider, J. B.: Measurement of stratus cloud and drizzle parameters in ASTEX with a Ka-band Doppler radar and a microwave radiometer, J. Atmos. Sci., 52, 2788–2799, 1995.

Frisch, A. S., Shupe, M., Djalalova, I., Feingold, G., and Poellot, M.: The retrieval of stratus cloud droplet effective radius with cloud radars, J. Atmos. Oceanic Technol., 19, 835–842, 2002.

Hogan, R. J., Gaussiat, N., and Illingworth, A. J.: Stratocumulus Liquid Water Content from Dual-Wavelength Radar, *J. Atmos. Oceanic Technol.*, 22, 1207–1218, https://doi.org/10.1175/JTECH1768.1, 2005.

Huang, D., Johnson, K., Liu, Y., and Wiscombe, W.: High resolution retrieval of liquid water vertical distributions using collocated Ka-band and W-band cloud radars, *Geophys. Res. Lett.*, 36, L24807, https://doi.org/10.1029/2009GL041364, 2009.

Intrieri, J. M., Shupe, M. D., Uttal, T., & McCarty, B. J.: An annual cycle of Arctic cloud characteristics observed by radar and lidar at SHEBA, Journal of Geophysical Research: Oceans, 107(C10), SHE-5, https://doi.org/10.1029/2000JC000423, 2002.

Jackson, R. C., McFarquhar, G. M., Korolev, A. V., Earle, M. E., Liu, P. S. K., Lawson, R. P., Brooks, S., Wolde, M., Laskin, A., and Freer, M.: The dependence of ice microphysics on aerosol concentration in arctic mixed-phase stratus clouds during ISDAC and M-PACE: AEROSOL EFFECTS ON ARCTIC STRATUS, J. Geophys. Res. Atmospheres, 117, n/a-n/a, https://doi.org/10.1029/2012JD017668, 2012.

Järvinen, E., Nehlert, F., Xu, G., Waitz, F., Mioche, G., Dupuy, R., Jourdan, O., and Schnaiter, M.: Investigating the vertical extent and short-wave radiative effects of the ice phase in Arctic summertime low-level clouds, Atmospheric Chem. Phys., 23, 7611–7633, https://doi.org/10.5194/acp-23-7611-2023, 2023.

Kang, L., Marchand, R. T., & Wood, R.: Stratocumulus precipitation properties over the Southern Ocean observed from aircraft during the SOCRATES campaign, Journal of Geophysical Research: Atmospheres, 129(6), e2023JD039831, https://doi.org/10.1029/2023JD039831, 2024.

Korolev, A., & Milbrandt, J.: How are mixed-phase clouds mixed?, Geophysical Research Letters, 49(18), e2022GL099578, https://doi.org/10.1029/2022GL099578, 2022.

Maciel, F. V., Diao, M., and Yang, C. A.: Partition between supercooled liquid droplets and ice crystals in mixed-phase clouds based on airborne in situ observations, Atmos. Meas. Tech., 17, 4843–4861, https://doi.org/10.5194/amt-17-4843-2024, 2024.

Mioche, G., Jourdan, O., Delanoë, J., Gourbeyre, C., Febvre, G., Dupuy, R., Monier, M., Szczap, F., Schwarzenboeck, A., and Gayet, J.-F.: Vertical distribution of microphysical properties of Arctic springtime low-level mixed-phase clouds over the Greenland and Norwegian seas, Atmos. Chem. Phys., 17, 12845–12869, https://doi.org/10.5194/acp-17-12845-2017, 2017.

NCAR/EOL HCR Team, NCAR/EOL HSRL Team: SOCRATES: NCAR HCR radar and HSRL lidar moments data. Version 3.2, UCAR/NCAR - Earth Observing Laboratory [data set], https://doi.org/10.5065/D64J0CZS, 2023.

Oh, S.-B., Lee, Y. H., Jeong, J.-H., Kim, Y.-H., and Joo, S.: Estimation of the liquid water content and Z–LWC relationship using Ka-band cloud radar and a microwave radiometer, Meteorological Applications, 25, 423–434, https://doi.org/10.1002/met.1710, 2018.

Romatschke, U., Dixon, M., Tsai, P., Loew, E., Vivekanandan, J., Emmett, J., and Rilling, R.: The NCAR Airborne 94-GHz Cloud Radar: Calibration and Data Processing, Data, 6, 66, https://doi.org/10.3390/data6060066, 2021.

Romatschke, U., Dixon, M. J.: Vertically Resolved Convective–Stratiform Echo-Type Identification and Convectivity Retrieval for Vertically Pointing Radars, J. Atmos. Oceanic Technol., 39, 1705–1716, https://doi.org/10.1175/JTECH-D-22-0019.1, 2022.

Romatschke, U., and Vivekanandan, J.: Cloud and precipitation particle identification using cloud radar and lidar measurements: Retrieval technique and validation, Earth and Space Science, 9, e2022EA002299, https://doi.org/10.1029/2022EA002299, 2022.

Schima, J., McFarquhar, G., Romatschke, U., Vivekanandan, J., D'Alessandro, J., Haggerty, J., et al.: Characterization of Southern Ocean Boundary Layer Clouds using airborne radar, lidar, and in situ cloud data: Results from SOCRATES, Journal of Geophysical Research: Atmospheres, 127, e2022JD037277, https://doi.org/10.1029/2022JD037277, 2022.

Shupe, M. D., Kollias, P., Matrosov, S. Y., and Schneider, T. L.: Deriving mixed-phase cloud properties from Doppler radar spectra, J. Atmos. Oceanic Technol., 21, 660–670, https://doi.org/10.1175/1520-0426(2004)021<0660>2.0.CO;2, 2004.

Shupe, M. D., Uttal, T., and Matrosov, S. Y.: Arctic Cloud Microphysics Retrievals from Surface-Based Remote Sensors at SHEBA, Journal of Applied Meteorology and Climatology, 44, 1544–1562, https://doi.org/10.1175/JAM2297.1, 2005.

Shupe, M. D.: A ground-based multisensor cloud phase classifier, Geophysical Research Letters, 34, L22809, https://doi.org/10.1029/2007GL031008, 2007.

Toledo, F., Haeffelin, M., Wærsted, E., and Dupont, J.-C.: A new conceptual model for adiabatic fog, Atmos. Chem. Phys., 21, 13099–13117, https://doi.org/10.5194/acp-21-13099-2021, 2021.

Tridon, F., A. Battaglia, P. Kollias, E. Luke, and C. R. Williams: Signal Postprocessing and Reflectivity Calibration of the Atmospheric Radiation Measurement Program 915-MHz Wind Profilers. J. Atmos. Oceanic Technol., 30, 1038–1054, https://doi.org/10.1175/JTECH-D-12-00146.1, 2013b

Wu, W., and McFarquhar, G.: NSF/NCAR GV HIAPER 2D-S Particle Size Distribution (PSD) Product Data. Version 1.1, UCAR/NCAR - Earth Observing Laboratory [data set], https://doi.org/10.26023/8HMG-WQP3-XA0X, 2019.

Xi, B., Dong, X., Zheng, X., and Wu, P.: Cloud phase and macrophysical properties over the Southern Ocean during the MARCUS field campaign, Atmospheric Measurement Techniques, 15, 3761–3777, https://doi.org/10.5194/amt-15-3761-2022, 2022.

Zaremba, T. J., Rauber, R. M., McFarquhar, G. M., Hayman, M., Finlon, J. A., and Stechman, D. M.: Phase characterization of cold sector Southern Ocean Cloud tops: Results from SOCRATES, Journal of Geophysical Research: Atmospheres, 125, e2020JD033673, https://doi.org/10.1029/2020JD033673, 2020.

Zheng, X., Dong, X., Xi, B., Logan, T., and Wang, Y.: Distinctive aerosol–cloud–precipitation interactions in marine boundary layer clouds from the ACE-ENA and SOCRATES aircraft field campaigns, Atmos. Chem. Phys., 24, 10323–10347, https://doi.org/10.5194/acp-24-10323-2024, 2024.

---

## Author Comment (AC2)

**Response to Reviewer:**

We thank and appreciate the reviewers for their valuable scrutiny and feedback on our manuscript and the opportunity to submit a revised version. All the suggested changes and feedback have been duly addressed and implemented, with our responses for each comment presented as follows.

The major changes are:

1. A supplementary material has been added alongside to the revised manuscript containing the validation of the estimated cloud base and LWP calculations.

2. The CFAD analysis is completely revised.

3. The wording of the manuscript has been significantly improved to remove redundancies and increase clarity. Additions have been made as per reviewer suggestions.

Note: line numbers must have changed in the revised manuscript.

**Reviewer 2 Comments:**

**1.** The authors should specify the purpose of this study, for example to identify cloud boundaries in the absence of lidar. In order to provide more clarity on the paper's objective.

**Response:** The objectives of the study are as mentioned in the introduction section: we aim to use a combination of both in-situ and radar-based measurements during SOCRATES to develop a new method to classify the MBL cloud phase and determine cloud boundaries over the SO for low-level clouds, to answer the following questions:

- What are the dominant cloud types, their associated cloud phase, base and top heights, and their vertical distribution?
- What are the phase-specific macrophysical properties for SO low-level clouds?

In this study, however, the radar + lidar signals has an overlap region of only ~11% (as observed from this study based on a total of 15 research flights) when considering the entire 3D time-height profiles where lidar attenuates easily along the vertical. Radar signals contribute to ~84% of the entire valid cloud signal retrievals (3D profiles) while lidar contributes to ~16% (this is based on the valid cloud reflectivity and backscatter measurements as defined by Shupe., 2007). However, along the time axis the overlap is almost 100% hence a direct comparison of our dominant phase results and Shupe (2005), Initrieri (2002) method based on median lidar PLDR (and Backscatter coefficient) applied to the SOCRATES dataset as in Table 4b was possible (since both are retrieved as 1D time-series profiles).

**2.** I would advise authors to estimate uncertainties more accurately, simply by adding statistical parameters (standard deviation on figures), for example. For in situ probes and parameters, refer to more articles highlighting measurement uncertainties and uncertainties related to derived parameters. Above all, to show that in situ accurate and real, but is nevertheless marred by uncertainties.

**Response:** The values presented in Table 3 and Table 5, along with the higher degree of spread of the frequencies in CFAD indicate the greater level of uncertainties that exists in the measurements during SOCRATES. Furthermore, the cited papers in Table 1 (reference column), also present these uncertainties in measurements to some level for each in-situ probe and remote sensing measurements, some of these cited studies in Table 1 are entirely on estimating the level of uncertainties of measurements from the respective instruments. The cited literatures throughout the paper on classifying cloud-type, cloud phase and hydrometeor-type detection over the SO region, including Xi

(2022), Desai (2023), D'Alessandro (2021, 2019), Romatschke & Vivekanandan (2022), Atlas (2021), Schima (2022), Zaremba (2020) also indicate the level of uncertainties existing for measurements over the SO. We do compare the levels of uncertainties with the results in this study with other similar studies like Xi (2022) and the uncertainties show decent agreement with previous studies. Even for a greater precision calculation of constrained height-resolved LWC as in Huang (2009) and Hogan (2005) using dual-frequency radars, the retrieved LWP still showed a mean difference of 70-120 $g/m^2$ with microwave radiometer (MWR) measured LWP.

**3.** The authors should explain the LWP calculation a little more clearly. This is an important point in the paper, linking in situ and phase discrimination. I would advise the authors to try to validate their methods cloud boundary detection and LWP : For LWP, they should try to compare the LWP calculation with a sawtooth portion of flight in a cloud. In order to see qualitatively whether the LWP is "globally" of the same order of magnitude. For cloud boundaries, you could once again use thresholds on your contents (LWC and/or IWC) to estimate cloud top and base on sawtooth leg (as in the portion on your Fig. 1 between 26 and 27 UTC for example).

**Response:** This was an excellent suggestion, and we appreciate the reviewer pointing this out. The LWP calculation is simply based on method used in existing studies like Oh (2018), Mioche (2017) which integrates the LWC across the cloud thickness. Validating the calculated LWP as in eq. 3 in the manuscript with the LWP measured across the sawtooth regions where the sampling height is the flight altitude and LWC is measured along that same height as well (i.e. LWP = LWC*flight_altitude for the sawtooth regions) shows excellent agreement. The correlation value (r) is 0.85 with a strong statistical significance (p<<0.001). Cloud Top values are consistent to the maximum height for the reflectivity profiles of the sampled clouds, this method seems very accurate for estimating LWP that also presented in Kang (2024). However, this method does not apply accurately for cloud base estimation, as only one or two data points at the extremities of the sawtooth legs are indicative of the base and top heights, resulting in very few comparable data points and makes the statistical comparison insignificant. We have tried to compare and validate our cloud base and LWP calculations with existing geographically and spatio-temporally collocated measurements during the MARCUS campaign which provided radiometer measured LWP values and, ceilometer and micropulse lidar estimated cloud base over the SO. We also compare the LWP and cloud base height values by creating a lat-lon grid box with overlapping measurements for low level clouds between the two campaigns SOCRATES and MARCUS for the similar climatological months (Jan, Feb and March), and found excellent agreement in the resulting values. The mean cloud base height difference was found to be around 159 meters where SOCRATES mean cloud base height (derived using the method described in this study) was 1055 meters and for MARCUS was ~900 meters. This is expected as MARCUS being a ship campaign sampled more lower clouds (closer to the surface) than SOCRATES, and the mean difference is within a decent agreeable range for cloud base heights. LWP measurements also show an excellent agreement with mean values: 110 $g/m^2$ (SOCRATES, as calculated in this study) and 127 $g/m^2$ (MARCUS, radiometer-measured), and a difference of ~17 $g/m^2$.

The comparisons and validations of cloud base heights and LWP measurements are presented in the Supplementary as will be attached with the revised manuscript.

**4.** The overall weather conditions for the campaign were not presented.

- ECMWF or ERA weather

I think it would be interesting to add a paragraph highlighting thermodynamic conditions.

In particular, to link with cloud structures and the strong presence of low-level clouds.

- Temperature to map cloud properties

**Response:** Currently this is beyond the scope of this study however we have added some lines describing the thermodynamic and weather conditions in the revised manuscript. Previous studies on the SOCRATES campaigns cited throughout the manuscript have already covered extensively on the thermodynamic conditions prevailing over the SO region during the austral summer months. The SOCRATES campaign specifically targeted low-level clouds with greater liquid concentrations hence our results that estimated a >90% occurrence of low-level clouds and liquid-dominant cloud phase was expected. The ERA5 temperature was used for the phase-analysis which was already available as a time-height profile matched to the HCR-HSRL datasets in the SOCRATES EOL data archive, developed by NCAR/EOL HCR Team (2023). SOCRATES campaign was carried out during the austral summer months of Jan and Feb 2018 hence there was not any seasonal differences observed between the datasets as the months of NDJFM present relatively stable and similar cloud cover over the SO (Dong et al., 2024). The linkages of temperature (entire cloud, cloud top and cloud base) with the retrieved cloud phases are presented in Figure 8. Maciel (2024) showed extensively on the cloud phase partition linkages with the thermodynamic temperatures (cited in the revised manuscript). Furthermore, a typically windy weather condition was observed along the cloud base, which is a characteristic of the SO, also indicated by the greater level of turbulence observed in the datasets captured by the profiles of reflectivity, spectrum width and doppler velocity (Figure 7) and the CFADs (Figure 10).

**5.** Part 4 (Results and discussion) is interesting, with a detailed phase classification algorithm. The sub-sections are coherent, and the addition of the comparison of the different methods is an essential and clear point. However, I find the explanations of the CFADs linked to figure 10. a little complex and difficult to follow. The links between Doppler velocity, or Spectrum Width, and size distributions are sometimes complicated to make. Moreover, add definition of mixed phase; please provide ref. It is important to clary what is considered as a mixed phase cloud or layer here. (see specific comment).

**Response:** We have added a definition for the mixed phase cloud as per Korelov (2022) in the revised manuscript (Section 4.1). We appreciate this comment as it helped us identify certain key issues in the CFAD presented, which led us to reanalyze the CFADs and the new CFAD is presented in the revised manuscript. The key changes are that since a direct pixel by pixel mapping of the phase profile was tampering the 2D time-height structure of the reflectivity, spectrum width and doppler velocity profiles. In the new CFAD we carefully remove all the surface values and only retain the profiles from the estimated cloud base height to the cloud top height eliminating near-surface contamination or noise. Furthermore, the time averaging is increased to 1 minutes from the previous 10 seconds to further smoothen the data to handle outliers and remove significant noise from the volatility of the remote sensed measurements. This helped to significantly smoothen and polish the dataset prior to plotting the CFADs. The revised CFAD highlight the dependence of dBZ on particle size, while $V_d$ and WID profiles reveal particle motion and turbulence, with findings showing turbulence-induced broadening near cloud bases and transitions in particle sizes and phases from cloud tops to bases. Mixed-phase and ice clouds show distinct patterns, with turbulence and particle growth playing critical roles in their vertical evolution. The new findings suggest a median dBZ of around -26 for liquid phase with a slight shift towards -30 from lower mid to cloud base (normalized height, $H_i<0.2$), dBZ for mixed-phase remains significantly high at around -20 dBZ for mixed phase displaying a greater spread which can be attributed to the wider variability in particle size distributions, ice phase displays a median dBZ of around -27 from cloud top to mid-cloud level but increases to -25 at the cloud base due to incidence of larger particle sizes as ice crystals grow from the cloud top to cloud base by accumulation and aggregation processes. The doppler velocity ($V_d$) shows similar trend across all the three phases increasing significantly from the mid to cloud base level ($H_i<0.2$) due to significant downwelling motion. The spectrum width (WID) of liquid and mixed phase cases are similar exhibiting sharp increase from $H_i<0.2$ indicative of significant downwelling motion and greater turbulence at the cloud base, for ice phase the median WID is significantly higher at the cloud top (0.7 m/s), decreases at mid cloud levels and again increases to ~0.6 m/s at cloud base which is indicative of a broader spread in the ice particle size distribution at the upper levels and significant turbulence and downwelling motion at the cloud base. The CFADs also confirm the irregular shape or morphology of ice particles along with the higher incidence of larger drizzle and ice particles at the cloud base, but not enough liquid samples

at the lower bottom regions. The results from Maciel (2024) have shown the presence of stronger in-cloud turbulence and updraughts in the clouds sampled during SOCRATES, especially for cases when supercooled liquid droplets are surrounded by ice crystals and/or mixed phase cases. This is demonstrated by the nature of the CFADs where a stronger turbulence causes an increase in Vd and WID values near the cloud base (Hi<0.2). Detailed analysis is presented in the revised manuscript.

**6.** You can make comparison with other study, it looks like a statistical comparison and not a pixel/cloud layer by pixel comparison (which could be more relevant). It could interesting to carry out a pixel by pixel phase comparison. Phase comparison with other in situ probes such as the PHIPS which can provide an independent assessment of the cloud phase. Provide in the appendix the results of the D'Alessandro method applied to your dataset.

**Response:** We did not identify any existing phase retrieval study that could offer a pixel-by-pixel comparison with this study. Hence an aggregate comparison is the best we could do here as is presented in Table 4a and 4b. PHIPS dataset do not provide a relevant phase-specific dataset that is comparable to our estimated time-height or dominant phase profiles, such comparisons though relevant are not apple-apple comparison. The presented comparison of our results with MLR algorithm in Table 4a is the results of D'Alessandro phase product applied to this dataset, we however cannot independently do what D'Alessandro method did as it was an extensive multinomial logistic regression analysis and beyond the scope of our study. We used the final cloud phase product from D'Alessandro (2021) method which is available at the SOCRATES EOL data archive for the comparisons with our study by matching the time stamps and found decent agreements (Table 4a). The significant difference between the methodologies in this study and D'Alessandro (2021) have already been highlighted in section 4.2.

**Specific comments and technical corrections**

**7. Line 18 :** Please and microphysical to "macrophysical properties " as the in situ measurements (CDP and 2DS) are also used to characterize the microphysical structure of the clouds

**Response:** Corrected as suggested

**8. Line 24-25 :** Please specify the temperature range when you mention higher temperature and lower temperatures

**Response:** Corrected as suggested (as it is a range, hence T>-2.5 $^{\circ}$C for liquid and <-2.5 $^{\circ}$C for ice).

**Introduction :**

**9. Line 33 :** In the beginning of this introduction, all that's missing is a latitude scale to define the limits of the Southern Ocean globally, not just for the campaign region.

**Response:** Corrected as suggested

**10. Line 47 :** The are other papers focusing on Arctic mixed-phase clouds such as Jackson et al., (2012), Järvinen et al., (2023) or Moser et al., (2023) that describe the structure of these clouds from in situ data.

**Response:** We have cited and referenced these studies appropriately in the revised manuscript.

**11. Line 50-52 :** Cloud you be more specific ? What do you mean by "most algorithms are tuned for specific …. climatic regions" ? Does it concern satellite retrieval algorithm where the cloud phase identification is performed priori to cloud property retrievals ?

**Response:** Refer to Shupe (2007) for clarification on this part. It concerns both satellite and ground-based retrievals of cloud properties where phase is estimated apriori as a necessary pre-requisite for cloud properties evaluation. We have slightly modified the writing in the revised manuscript to make it clearer.

**12. Line 57 :** For SO another recent paper Bazantay et al., (2024) and for Arctic region there is also Mioche et al., (2015) or Matus and L'Ecuyer, (2017).

**Response:** Cited and referenced these studies appropriately in the revised manuscript.

**13. Line 79 :** In 1-2 sentences, the authors could propose an example of cloud-type classification based on satellite instruments. In order to get an overall idea of the different methods (in situ, spatial,…). Additionally, the authors should also state that the mixed phase identification depend on the observation scale. In the literature, different approach are used and for instance a mixed phase cloud can be composed of a combination of liquid phase pixels and ice phase clouds with no mixed phase pixels. It also depend of the instrument used to detect the cloud phase layers.

**Response:** We have tried our best to add couple lines addressing this in the revised manuscript, studies like Korelov (2022), D'Alessandro (2021) have been cited which cover the phase- and platform-specific information extensively especially for mixed phase partitioning in SO and Arctic clouds and for comparisons between ground, airborne and satellite-based measurements Dong (2024) can be referred. These information has been added suitably in Section 1 (Introduction) and Section 4.2 where they best fit.

**14. Line 82 :** This sentence is really long.

**Response:** Corrected as suggested.

**15. Line 88 :** Perhaps add instrument names for remote sensing as in line 92 for in situ probes.

**Response:** Corrected as suggested.

**16. Line 104 :** Authors can also add "and near-surface contamination problems related to echo".

**Response:** Corrected as suggested.

**17. Line 107 :** However, lidar is still useful for determining cloud tops if they are made up of liquid phase (strong backscattering).

**Response:** Our analysis showed that while HSRL lidar backscatter signal was efficient for estimating cloud base but only when the flight flew below the cloud base offering a zenith-viewing direction for the lidar, which is also presented in Kang (2024), but not for estimating cloud tops at those same zenith-viewing direction as the lidar signal attenuates for thicker cloud layers. However, the HCR radar reflectivity and spectrum width profiles were found to be comprehensive enough for estimating both cloud top and cloud base height at both the zenith and nadir-pointing directions. Similarly cloud base estimation using lidar at the nadir pointed view was found to be erroneous and we estimated an offset of ~400 meters between the cloud base estimated by the two instruments (HCR and HSRL).

**Data and methods :**

**18. Line 131 :** The authors should also state that the mixed phase identification depend on the observation scale. In the literature, different approach are used and for instance a mixed phase cloud

can be composed of a combination of liquid phase pixels and ice phase clouds with no mixed phase pixels. It also depend of the instrument used to detect the cloud phase layers.

**Response:** See response to point 13. Additionally, a small definition of mixed phase cases along with the type of mixed-phase partitioning observed during SOCRATES has been added in Section 4.2 as per Korelov (2022); Maciel et al., 2024; and D'Alessandro (2021), respectively. The subjectivity of phase retrieval to observational scales and nature of sampling is duly acknowledged as suggested (Section 1) however for this study the observational scales and nature of sampling are very specific to the SOCRATES campaign.

**19. Line 137 :** I find this sentence not very clear, the authors could add a "small" additional explanation to explain the liquid water/ice discrimination.

**Response:** Improved the writing as suggested. That's a screening criteria as per the 2DS instrumentation details for SOCRATES, that only particle sizes D>200 µm are considered to be ice. Explained in Table 1 as well.

**20. Line 140 :** The uncertainties arising from the in situ instruments are reflected in the PSDs, and I know that it's complicated to estimate uncertainties in secondary parameters (such as content (IWC or LWC)). Perhaps add a sentence to effect that these primary uncertainties are reflected in the secondary parameters.

**Response:** Corrected as suggested.

**21. Line 141 :** Do you use a weighted average to calculate your merged size distribution ?

**Response:** They are a continuous dataset at 1 second time interval where CDP contains PSDs for particles between 2-50 µm, and 2DS provides PSD for particles from 40 µm onwards (2DS does provide PSD from 10 µm but the dataset suggests using diameter > 40 um as sizes below that cannot be resolved well). We use the CDP and 2DS dataset as it is, without any modifications, merging them as a continuous size distribution following Zheng (2024). Refer Figure 2 and Zheng (2024), the droplet number concentrations in the overlapping size bin between CDP and 2DS are redistributed, assuming a gamma distribution, and thereby a complete size spectrum of cloud and drizzle can be merged from CDP and 2DS measurements.

**22. Line 142** : How did you choose the threshold (40 µm) between cloud droplets and drizzle particles ?

**Response:** The demarcation was selected based on Wood (2005) and Zheng (2024). The CDP can only resolve for smaller cloud droplets (2-50 µm) while 2DS can sample drizzle particles as well. 2DS dataset mentions that size distributions cannot be resolved for particles below 40 µm. (Refer: Wu and McFarquhar., 2019). Also see: https://data.eol.ucar.edu/file/download/5406994F0D471/Readme_2DS_V1.1.txt

**23. Line 152 :** Out of curiosity, did you check whether the ERA5 data matched the temperature data measured by the aircraft ?

**Response:** It is difficult to directly compare them but for segments where the ERA5 height (radar height) is approximately at the flight altitude level, ERA5 and aircraft temperatures had excellent agreements. But we did not scrutinize it in much detail as this is an already published dataset for the SOCRATES campaign. Refer: NCAR/EOL HCR Team, NCAR/EOL GV-HSRL Team. 2023. SOCRATES: NCAR HCR radar and GV-HSRL lidar moments data. Version 3.2. UCAR/NCAR - Earth Observing Laboratory. https://doi.org/10.5065/D64J0CZS. Accessed 11 Nov 2024.

**24. Line 197 :** The time scale is a bit strange (29 hours UTC ? ), maybe change the legend to "Since midnight (15 Jan 2018)".

**Response:** The time axis has been corrected in the revised manuscript and converted to have the corrected UTC decimal time.

**25. Line 230 :** I'm not sure I understand the LWP calculation. In the formula, j corresponds to ? In agreement with *h et al. (2018)*, j represents the number of points in your profile ? To have several values of LWC in the profile, means that the plane passes several times in the same column ? Or is your in situ LWC just summed over the thickness of your cloud ? This calculation and method should be described a little more, as it is essential for the study. What does n stand for ? Authors can attempt to "validate" the LWP, using the method in Mioche et al., (2017), on a sawtooth leg.

**Response:** The LWP calculation is actually done as per Oh (2018), and Mioche (2017) (also see Toledo., 2021). The j is each time interval (1 sec), which is the time frequency for the sampling. The flight sampling for measuring PSD and LWC is along the straight track, it does not pass several times in the same column. n is the last extreme sampling point (upper limit), i.e. where the flight sampling leg ends. i, j and n are all indices or points along the time dimension. In-situ LWC is not summed across the cloud thickness but is directly measured along the flying altitude or the sampling height. SOCRATES do not provide time-height profiles for LWC or LWP as has been mentioned in the manuscript. (note: LWC is also directly available from 2DS and CDP measurements and is validated to be same as the calculated merged LWC, for respective size bins). LWP validation has been done (refer to response to comment 3) and details has been added to the Supplementary.

**26. Line 250 :** How is the IWC calculated ? Mass/diameter law?

**Response:** We do not calculate IWC ourselves and use it directly from the 2DS dataset, where IWC is estimated using the mass-diameter relationship. (Refer: Wu, W., McFarquhar, G. 2019. NSF/NCAR GV HIAPER 2D-S Particle Size Distribution (PSD) Product Data. Version 1.1. UCAR/NCAR - Earth Observing Laboratory. https://doi.org/10.26023/8HMG-WQP3-XA0X).

**27. Line 253 :** How good is the ERA5 data in the Southern Ocean ?

**Response:** We only use the ERA5 temperature in this study and we found it to be in-sync with the in-situ aircraft measured temperature, at the similar height levels. However, a direct comparison is not possible. The existing literature on SOCRATES can be referred here for further details, like Romatschke (2021); Romatschke and Dixon (2022); Romatschke and Vivekanandan (2022). Further details on the dataset can be found in NCAR/EOL HCR Team, NCAR/EOL GV-HSRL Team (2023).

**28. Line 267 :** The authors could add the grids on the figures to make it easier to see the values (even if the essential values are quoted in the text). I think a error bar should be added to figs 3.a, 4.a and b. Ilt's always interesting to estimate the statistical error.

**Response:** Gridlines have been added to Fig 3, 4 in the revised manuscript. The error statistics is mentioned in Table 3. We will add the error bars in Fig 4a in the revised manuscript, it is around 10 $g/m^2$ (standard error) as described in the manuscript already. The estimated LWP was also compared with reported LWP measurements over the SO for low-clouds, and we found a decent agreement in the values and the uncertainty (std. dev) with results from Xi (2022), Mace (2023), Tan (2023). See Xi (2022) - Figures 3 and 4 which is a similar version of Fig. 3 and 4 of this study. Figures 3a and 4b are occurrence frequency percentages and the occurrence frequencies add up to 100% for all cloud types.

**29. Line 271 :** I would have preferred the definition of the LWP threshold at 10 g/m² to have been explained at the same time. This is explained in the next section (4.1).

**Response:** Since the LWP thresholding is only carried prior to phase estimation, it was deemed more appropriate to be explained in that section (4.1). We will add it briefly in the referenced section as well as mentioned.

**30. Line 285 :** Have you analyzed median values versus mean values ? Just to see if the observations follow a Gaussian curve.

**Response:** We have for this part, however not a perfect Gaussian curve, the mean and median values are very similar to each other for the cloud boundaries and LWP. However, the CFADs (Figure 10) were found to be slightly right skewed.

**Results and discussions :**

**31. Line 291 :** 10 seconds at ± 100 m/s (probably more) ≈ 1 km. It is important to note that the main underlying hypothesis is that the cloud is globally homogeneous over ± 1 km.

**Response:** Correctly identified, and we have added this to the revised manuscript. The temporal averaging is an important constraint in this study. The homogenous nature of SO clouds has already been highlighted in the Introduction section as well.

**32. Line 340 :** Duplicate sentence
**Response:** Corrected.

**33. Line 352 :** In the fig 6.a, the pixels representing the point counts are wider than in fig 6.d. What is the reason for this difference in definition ? Is it just that there are fewer points for these conditions ("liquid cloud droplets, drizzle and rain drops") ?

**Response:** Correct, number of points for $T > 0\ ^{\circ}C$ is very limited hence a lower number of data bins (50 bins only) was selected for constructing Fig 6a for enhancing visual clarity, resulting in wider pixels.

**34. Line 390 :** I find this paragraph difficult to understand, but it's important for understanding dimensional segmentation. The authors could perhaps be reworked.

**Response:** Reworded this section as suggested.

**35. Line 432 :** It would have been interesting to have 2DS images to represent the morphological environment. But we can't show everything.

**Response:** We did try to analyze some of the 2DS particle habit imager images with our retrieved phase results. We found some decent agreement, but an apple-to-apple comparison was not seen possible due to the nature of the habit imager dataset (and our temporal averaging). The 2DS habit image data can be found here (UCAR/NCAR - Earth Observing Laboratory. 2018. NSF/NCAR GV HIAPER Raw 2D-S Imagery. Version 1.1. UCAR/NCAR - Earth Observing Laboratory. https://doi.org/10.26023/9555-DKY0-J604.) and requires the XPMS2D software to be viewed.

**36. Line 433 :** The authors could add a reference to show the consistency of these remarks with other studies, also in agreement with mixed-phase clouds in the Arctic region. I also think that the PHIPS and SID-3 instruments were deployed during the SOCRATES campaign. Did you check the consistency of your cloud phase identification with asymmetry parameter derived from the PHIPS which could be a good proxy for cloud phase. SID-3 can also provide information on small ice crystals. Was this investigated?

**Response:** See response to comment 6. Since we were specific to low-level SO clouds during SOCRATES however Korelov (2022) which is one prominent study on arctic mixed phase clouds which was thoroughly cited throughout the paper, we did not carry out any further comparisons with Arctic region clouds. As, mentioned, PHIPS habit imager dataset does not provide a relevant phase-specific dataset that is comparable to our estimated time-height or dominant phase profiles, such comparisons though relevant are not apple-apple. We could not find any SID-3 dataset deployed during SOCRATES in the EOL archive (this needs to be requested from the relevant research team as per Järvinen, 2018). So, this was not investigated, but a future study can be designed to validate these findings. One must note that our proposed phase estimation method is coarser and majorly dependent on radar signals, hence comparisons with particle habits is not straightforward.

**37. Line 435 :** So yes, the 2DS is better to identifying images above 50 μm, but the term "easily" is a little misleading. Particle with size of 50 μm is made up of ± 5 pixels with 2DS, so it's still difficult to characterize the phase and even more complicated if you're trying to analyse morphology.

**Response:** Yes, this is true. We have reworded the sentence. Also noticed a typo in this section as ice particles only above 200 um is resolvable clearly.

**38. Line 454/503 :** Authors could try to explain the difference between the comparison of the 2 methods (this study and MLR). Of course, these differences can be partly explained by the use of very different methods.

**Response:** A brief description of the MLR algorithm has already been presented in the manuscript, which is a readily available cloud phase product dataset in the SOCRATES EOL data archive, that is also indicative of how the methods are different in nature. As for Shupe (2005) and Initrieri (2002) we used those thresholds and constraints on our dataset to estimate phase according to their methods to compare with our results. Further, see response to comment 6.

**39. Line 480 :** What could account for the difference in PLDR thresholds between Arctic and Southern Ocean ?

**Response:** Majorly the nature of the clouds as described in the Introduction section, which also accounts for the different microphysical nature, spatial heterogeneity, and the nature of ice-liquid phase partitioning in SO clouds compared to MBL clouds over the Arctic region. But as observed the PLDR thresholds we propose based on our findings in Section 4.2 are very close if not at all same to those defined over the Arctic. Further scrutiny is necessary.

**40. Line 535 :** What's not necessarily explained is that these thermodynamic parameters are dependent on the season in which the measurement campaign took place (January/February). Would partitioning phase or phase distribution be the same for identical environmental conditions in summer?

**Response:** We have added a sentence highlighting the exclusivity of these thermodynamic parameters to Jan/Feb (during SOCRATES). The results should be similar for the entire climatological winter months NDJFM (Dong, 2024) but further scrutiny is needed to evaluate how they change for the summer months.

**41. Line 571 :** I would have liked to see a reference for the influence of morphology on reflectivity unless this is a hypothesis you're proposing ?

**Response:** The CFAD analysis has been revised significantly, and the new analytical results and explanations need to be considered here. One key takeaway is that mixed phase which constitutes of intermixed small to large sized liquid and ice droplets can significantly cause higher and a greater spread in reflectivity values. Xi (2022) is a relevant reference to validate this finding.

**REFERENCES:**

Atlas, R., Mohrmann, J., Finlon, J., Lu, J., Hsiao, I., Wood, R., & Diao, M.: The University of Washington Ice-Liquid Discriminator (UWILD) improves single-particle phase classifications of hydrometeors within Southern Ocean clouds using machine learning, Atmospheric Measurement Techniques, 14(11), 7079–7101, https://doi.org/10.5194/amt-14-7079-2021, 2021.

Bazantay, C., Jourdan, O., Mioche, G., Uitz, J., Dziduch, A., Delanoë, J., Cazenave, Q., Sauzède, R., Protat, A., and Sellegri, K.: Relating Ocean Biogeochemistry and Low-Level Cloud Properties Over the Southern Oceans, Geophys. Res. Lett., 51, e2024GL108309, https://doi.org/10.1029/2024GL108309, 2024.

Chandra, A., Zhang, C., Kollias, P., Matrosov, S., and Szyrmer, W.: Automated rain rate estimates using the Ka-band ARM zenith radar (KAZR), Atmos. Meas. Tech., 8, 3685–3699, https://doi.org/10.5194/amt-8-3685-2015, 2015.

D'Alessandro, J. J., Diao, M., Wu, C., Liu, X., Jensen, J. B., & Stephens, B. B.: Cloud phase and relative humidity distributions over the Southern Ocean in austral summer based on in situ observations and CAM5 simulations, Journal of Climate, 32(10), 2781–2805, https://doi.org/10.1175/JCLI-D-18-0232.1, 2019.

D'Alessandro, J. J., McFarquhar, G. M., Wu, W., Stith, J. L., Jensen, J. B., & Rauber, R. M.: Characterizing the Occurrence and Spatial Heterogeneity of Liquid, Ice, and Mixed Phase Low-Level Clouds Over the Southern Ocean Using in Situ Observations Acquired During SOCRATES, Journal of Geophysical Research: Atmospheres, 126(11), https://doi.org/10.1029/2020JD034482, 2021.

D'Alessandro, J., Schima, J., McFarquhar, G.: SOCRATES Cloud Phase Product. Version 1.0. UCAR/NCAR - Earth Observing Laboratory [data set], https://doi.org/10.26023/S6WS-G5QE-H113, 2022.

Desai, N., Diao, M., Shi, Y., Liu, X., & Silber, I.: Ship-Based Observations and Climate Model Simulations of Cloud Phase Over the Southern Ocean, Journal of Geophysical Research: Atmospheres, 128(11), https://doi.org/10.1029/2023jd038581, 2023.

Dong, X., Das, A., Xi, B., Zheng, X., Behrangi, A., Marcovecchio, A. R., and Girone, D. J.: Quantifying the differences in Southern Ocean clouds observed by radar and lidar from three platforms, *Geophys. Res. Lett.*, [in review], 2024.

Hogan, R. J., Gaussiat, N., and Illingworth, A. J.: Stratocumulus Liquid Water Content from Dual-Wavelength Radar, *J. Atmos. Oceanic Technol.*, 22, 1207–1218, https://doi.org/10.1175/JTECH1768.1, 2005.

Huang, D., Johnson, K., Liu, Y., and Wiscombe, W.: High resolution retrieval of liquid water vertical distributions using collocated Ka-band and W-band cloud radars, *Geophys. Res. Lett.*, 36, L24807, https://doi.org/10.1029/2009GL041364, 2009.

Intrieri, J. M., Shupe, M. D., Uttal, T., & McCarty, B. J.: An annual cycle of Arctic cloud characteristics observed by radar and lidar at SHEBA, Journal of Geophysical Research: Oceans, 107(C10), SHE-5, https://doi.org/10.1029/2000JC000423, 2002.

Jackson, R. C., McFarquhar, G. M., Korolev, A. V., Earle, M. E., Liu, P. S. K., Lawson, R. P., Brooks, S., Wolde, M., Laskin, A., and Freer, M.: The dependence of ice microphysics on aerosol concentration

in arctic mixed-phase stratus clouds during ISDAC and M-PACE: AEROSOL EFFECTS ON ARCTIC STRATUS, J. Geophys. Res. Atmospheres, 117, n/a-n/a, https://doi.org/10.1029/2012JD017668, 2012.

Järvinen, E., Nehlert, F., Xu, G., Waitz, F., Mioche, G., Dupuy, R., Jourdan, O., and Schnaiter, M.: Investigating the vertical extent and short-wave radiative effects of the ice phase in Arctic summertime low-level clouds, Atmospheric Chem. Phys., 23, 7611–7633, https://doi.org/10.5194/acp-23-7611-2023, 2023.

Kang, L., Marchand, R. T., & Wood, R.: Stratocumulus precipitation properties over the Southern Ocean observed from aircraft during the SOCRATES campaign, Journal of Geophysical Research: Atmospheres, 129(6), e2023JD039831, https://doi.org/10.1029/2023JD039831, 2024.

Korolev, A., & Milbrandt, J.: How are mixed-phase clouds mixed?, Geophysical Research Letters, 49(18), e2022GL099578, https://doi.org/10.1029/2022GL099578, 2022.

Maciel, F. V., Diao, M., and Yang, C. A.: Partition between supercooled liquid droplets and ice crystals in mixed-phase clouds based on airborne in situ observations, Atmos. Meas. Tech., 17, 4843–4861, https://doi.org/10.5194/amt-17-4843-2024, 2024.

Mioche, G., Jourdan, O., Delanoë, J., Gourbeyre, C., Febvre, G., Dupuy, R., Monier, M., Szczap, F., Schwarzenboeck, A., and Gayet, J.-F.: Vertical distribution of microphysical properties of Arctic springtime low-level mixed-phase clouds over the Greenland and Norwegian seas, Atmos. Chem. Phys., 17, 12845–12869, https://doi.org/10.5194/acp-17-12845-2017, 2017.

NCAR/EOL HCR Team, NCAR/EOL HSRL Team: SOCRATES: NCAR HCR radar and HSRL lidar moments data. Version 3.2, UCAR/NCAR - Earth Observing Laboratory [data set], https://doi.org/10.5065/D64J0CZS, 2023.

Oh, S.-B., Lee, Y. H., Jeong, J.-H., Kim, Y.-H., and Joo, S.: Estimation of the liquid water content and Z–LWC relationship using Ka-band cloud radar and a microwave radiometer, Meteorological Applications, 25, 423–434, https://doi.org/10.1002/met.1710, 2018.

Romatschke, U., Dixon, M., Tsai, P., Loew, E., Vivekanandan, J., Emmett, J., and Rilling, R.: The NCAR Airborne 94-GHz Cloud Radar: Calibration and Data Processing, Data, 6, 66, https://doi.org/10.3390/data6060066, 2021.

Romatschke, U., Dixon, M. J.: Vertically Resolved Convective–Stratiform Echo-Type Identification and Convectivity Retrieval for Vertically Pointing Radars, J. Atmos. Oceanic Technol., 39, 1705–1716, https://doi.org/10.1175/JTECH-D-22-0019.1, 2022.

Romatschke, U., and Vivekanandan, J.: Cloud and precipitation particle identification using cloud radar and lidar measurements: Retrieval technique and validation, Earth and Space Science, 9, e2022EA002299, https://doi.org/10.1029/2022EA002299, 2022.

Schima, J., McFarquhar, G., Romatschke, U., Vivekanandan, J., D'Alessandro, J., Haggerty, J., et al.: Characterization of Southern Ocean Boundary Layer Clouds using airborne radar, lidar, and in situ cloud data: Results from SOCRATES, Journal of Geophysical Research: Atmospheres, 127, e2022JD037277, https://doi.org/10.1029/2022JD037277, 2022.

Shupe, M. D., Uttal, T., and Matrosov, S. Y.: Arctic Cloud Microphysics Retrievals from Surface-Based Remote Sensors at SHEBA, Journal of Applied Meteorology and Climatology, 44, 1544–1562, https://doi.org/10.1175/JAM2297.1, 2005.

Shupe, M. D.: A ground-based multisensor cloud phase classifier, Geophysical Research Letters, 34, L22809, https://doi.org/10.1029/2007GL031008, 2007.

Toledo, F., Haeffelin, M., Wærsted, E., and Dupont, J.-C.: A new conceptual model for adiabatic fog, Atmos. Chem. Phys., 21, 13099–13117, https://doi.org/10.5194/acp-21-13099-2021, 2021.

Wu, W., and McFarquhar, G.: NSF/NCAR GV HIAPER 2D-S Particle Size Distribution (PSD) Product Data. Version 1.1, UCAR/NCAR - Earth Observing Laboratory [data set], https://doi.org/10.26023/8HMG-WQP3-XA0X, 2019.

Xi, B., Dong, X., Zheng, X., and Wu, P.: Cloud phase and macrophysical properties over the Southern Ocean during the MARCUS field campaign, Atmospheric Measurement Techniques, 15, 3761–3777, https://doi.org/10.5194/amt-15-3761-2022, 2022.

Zaremba, T. J., Rauber, R. M., McFarquhar, G. M., Hayman, M., Finlon, J. A., and Stechman, D. M.: Phase characterization of cold sector Southern Ocean Cloud tops: Results from SOCRATES, Journal of Geophysical Research: Atmospheres, 125, e2020JD033673, https://doi.org/10.1029/2020JD033673, 2020.

Zheng, X., Dong, X., Xi, B., Logan, T., and Wang, Y.: Distinctive aerosol–cloud–precipitation interactions in marine boundary layer clouds from the ACE-ENA and SOCRATES aircraft field campaigns, Atmos. Chem. Phys., 24, 10323–10347, https://doi.org/10.5194/acp-24-10323-2024, 2024.